# The interplay between body mass index, motivation for food consumption, and noncommunicable diseases in the European population: A cross-sectional study

Marija Ljubičić[1], Marijana Matek Sarić[1*], Tamara Sorić[2], Ana Sarić[3], Ivo Klarin[1,4], Boris Dželalija[1], Alan Medić[1,5], Ivo Dilber[4], Ivana Rumbak[6], Jasmina Ranilović[7], Maria Papageorgiou[8], Viktória Szűcs[9], Elena Vittadini[10], Dace Klava[11], Lucia Frez Muñoz[12], Małgorzata Korzeniowska[13], Monica Tarcea[14], Ilija Djekić[15], Maša Černelič Bizjak[16], Raquel Guiné[17]

1 Department of Health Studies, University of Zadar, Zadar, Croatia, 2 Psychiatric Hospital Ugljan, Ugljan, Croatia, 3 Catholic University of Zagreb, School of Medicine, Zagreb, Croatia, 4 General Hospital Zadar, Zadar, Croatia, 5 Institute of Public Health Zadar, Zadar, Croatia, 6 Faculty of Food Technology and Biotechnology, Laboratory for Nutrition Science, Department of Food Quality Control, University of Zagreb, Zagreb, Croatia, 7 Research & Development, Podravka d.d., Koprivnica, Croatia, 8 Alexander Technological Educational Institute of Thessaloniki (ATEITh), School of Agriculture Technology, Food Technology and Nutrition, Department of Food Technology, Thessaloniki, Greece, 9 Hungarian Chamber of Agriculture, Directorate of Food Industry, Budapest, Hungary, 10 Department of Food Science, University of Parma, Parma, Italy, 11 Faculty of Food Technology, Department of Food Technology, Latvia University of Agriculture, Jelgava, Latvia, 12 The Netherlands Food Quality and Design Group, Wageningen University & Research, Wageningen, The Netherlands, 13 Faculty of Food Science, Department of Animal Products Technology and Quality Management Chełmońskiego, Wroclaw University of Environmental and Life Sciences, Wrocław, Poland, 14 Faculty of Medicine, Department of Community Nutrition and Food Safety, University of Medicine and Pharmacy Tirgu-Mures, Tirgu Mures city, Romania, 15 Faculty of Agriculture, Department of Food Safety and Quality Management, University of Belgrade, Zemun, Belgrade, Republic of Serbia, 16 Faculty of Health Sciences, Department of Nutritional Counselling-Dietetics, University of Primorska, Izola, Slovenia, 17 CERNAS-IPV, Polytechnic University of Viseu, Campus Politécnico, Repeses, Viseu, Portugal

* marsaric@unizd.hr

## Abstract

### Introduction

Consuming unhealthy foods in emotional states can increase body mass index (BMI), contribute to becoming overweight, and lead to the development of chronic diseases. This study aims to investigate the associations between BMI, emotional motivation for food consumption, and health outcomes.

### Materials and methods

"The Motivations for Food Choices" (EATMOT) questionnaire was used to assess the emotional components of food consumption in 9,036 individuals from 12 European countries. The multivariate analysis included linear and logistic regression to examine associations between variables.

---

**Data availability statement:** All relevant data are within the paper and its Supporting Information files.

**Funding:** The author(s) received no specific funding for this work.

**Competing interests:** The authors have declared that no competing interests exist.

## Results

Regression models confirmed associations between BMI, emotional motivation for food consumption (β = 0.13; p < 0.001), obesity (β = 0.35; p < 0.001), diabetes mellitus, and hypertension (β = 0.04; p < 0.001 for both). Using food as a coping mechanism for stress contributed to an increase in BMI [OR = 1.31 (95% CI 1.14–1.51); p < 0.001]. Emotional consolation was associated with a higher likelihood of an increased BMI [OR = 1.22 (95% CI 1.03–1.44); p = 0.020] and obesity [OR = 1.47 (95% CI 1.06–2.06); p = 0.022]. Participants with obesity had a greater likelihood of developing noncommunicable diseases, such as cardiovascular diseases [OR = 2.18 (95% CI 1.45–3.28); p < 0.001], diabetes mellitus [OR = 2.02 (95% CI 1.31–3.12); p = 0.001], hypercholesterolemia [OR = 1.62 (95% CI 1.13–2.32); p = 0.009], hypertension [OR = 1.85 (95% CI 1.36–2.52); p < 0.001], and gastric disorders [OR = 1.81 (95% CI 1.16–2.85); p = 0.010].

## Conclusion

These results underscore the need for targeted public health interventions that address emotional eating behaviors and promote healthier coping strategies to mitigate the risk of obesity and related health complications.

## Introduction

Chronic noncommunicable diseases (NCDs) are the leading cause of death and a daily threat to millions of people worldwide [1,2]. NCDs arise from a combination of genetic, environmental, and lifestyle factors. Modern life is often filled with tension, and contemporary lifestyles are recognized as significant contributors to NCDs [3]. Numerous studies show that maintaining a healthy lifestyle, including a balanced diet, regular physical activity, and smoking cessation, can prevent almost 80% of chronic diseases and premature deaths [3–5]. Despite the well-known relationship between NCDs, stress, and unhealthy lifestyle, many people are reluctant to change their behaviors to improve their health [6]. In response to stress and negative emotions, many individuals turn to food for comfort, often choosing energy-dense options rich in sugar, unhealthy fats, and salt [7]. Poor dietary choices, chronic stress, and insufficient physical activity all contribute to weight gain and the rising prevalence of obesity [8].

The prevalence of obesity has increased significantly over the last three decades [9]. Individuals with obesity have a higher risk of developing cardiovascular diseases (CVDs), type 2 diabetes mellitus (DM), fatty liver disease, neurodegenerative diseases, and cancer [9]. Although an unhealthy lifestyle increases body mass index (BMI) and contributes to the development of obesity [8], other factors can also play a role [9]. Obesity is a complex, multifactorial disorder influenced by biological, genetic, psychological, and environmental factors. Its biological and genetic basis encompasses a wide range of processes, including appetite regulation, metabolism,

neurological responses to food, and psychological responses to external and internal stimuli [9,10]. These complex mechanisms highlight the importance of standardized measures, such as BMI, not only for assessing and classifying body weight but also as a key tool for evaluating the risk of NCDs [11]. BMI is the most commonly used metric for identifying overweight and obesity [12,13]. Research has focused on the relationship between BMI and health-related behaviors [13], revealing that a high BMI correlates with increased fat consumption, low physical activity levels, excessive sedentary leisure time, and other unhealthy behaviors [7].

Studies show that overweight and obesity have reached pandemic proportions, posing major public health concerns [1,2,14]. The inflammatory components of obesity are strongly linked to major chronic diseases, both directly and indirectly [14]. For instance, obesity significantly increases the risk of NCDs such as type 2 DM, fatty liver disease, hypertension, myocardial infarction, stroke, osteoarthritis, Alzheimer's disease, depression, respiratory diseases, and certain cancers (including colon, liver, breast, prostate, ovarian, and kidney cancer). This leads to numerous socioeconomic challenges, including loss of productivity, unemployment, and a lower quality of life associated with disease [1].

The importance of BMI and the perceived value of health have garnered considerable research attention. However, few studies have thoroughly examined how factors such as motivation for health behavior (MHB) and emotional motivation for food consumption (EMFC), combined with BMI, interact with human health [6,7,15]. Emotional factors such as stress, anxiety, boredom, and depression can induce food consumption, increasing the tendency to use food as a coping mechanism for negative emotions [16]. Overwhelming and uncontrolled food consumption, influenced by genetic predispositions, biological appetite regulation, and psychological factors, can lead to the overconsumption of calorie-dense, nutrient-poor foods [9,10,17]. These negative tendencies of EMFC can have adverse health consequences, contributing to weight gain and obesity over time [16].

Conversely, maintaining health requires motivation and positive lifestyle choices, including a balanced diet, regular exercise, stress management, and avoiding smoking and alcohol [5,18]. Although lifestyle factors and MHB may be partly responsible for the occurrence of obesity, genetic variations also contribute significantly to this problem and its associated diseases [19]. Hence, it is important to recognize that the development of obesity is not only due to individual behaviors or a lack of motivation, as genetic, epigenetic, neurobiological, and environmental factors also play an important role in weight gain and its effects on different individuals [9,10,20]. However, MHB is complex and can vary widely across countries and regions due to cultural, social, economic, and environmental factors [15]. For all the aforementioned reasons, BMI, along with MHB and EMFC, is a crucial factor in preventing numerous NCDs [8].

There is a clear need to elucidate the mechanisms behind the relationship between emotions, food intake, motivation, and health outcomes, such as NCDs. The present study aims to assess the association between BMI, EMFC, MHB, and NCDs. We hypothesize that a higher BMI is associated with lower MHB and more pronounced EMFC.

Additionally, we assume that these factors are associated with an increased risk of being overweight and a higher incidence of critical NCDs, such as CVDs, DM, hypercholesterolemia, hypertension, gastric disorders (GDs), intestinal diseases (IDs), and obesity. While the focus of this study is on cross-national comparisons, it provides additional context for understanding how national differences in lifestyle, policy, and socioeconomic conditions influence BMI and the prevalence of NCDs. Early identification of emotionally unhealthy lifestyles and behaviors can positively impact awareness of the need to change these patterns. This can increase the consumption of healthy foods and ultimately improve physical and mental health, as well as overall well-being, thereby preventing the occurrence of NCDs.

## Materials and methods

This cross-sectional study is part of the multinational project EATMOT, led by the CI&DETS Research Centre of the Polytechnic Institute of Viseu, Portugal (PROJ/CI&DETS/2016/0008: EATMOT).

Some parts of the study design and methodology have already been described in previously published articles by the same research group [8,12]; thus, only a summary of these elements is provided in the following sections.

## Study design and participants

This cross-sectional study included 9,052 subjects from 12 European nations: Croatia (1,538; 17.0%), Greece (498; 5.5%), Poland (586; 6.5%), Portugal (1,314; 14.5%), Romania (821; 9.1%), Serbia (498; 5.5%), Hungary (500; 5.5%), Italy (541; 6.0%), Latvia (636; 7.0%), Lithuania (507; 5.6%), the Netherlands (521; 5.8%), and Slovenia (1,093; 12.1%). Participants were recruited from universities, schools, booths in shopping centers and marketplaces, downtown areas, rural regions, and other locations through advertisements or word-of-mouth. This recruitment method aimed to include participants of both genders, from urban and rural areas, and with diverse ages and educational backgrounds. The inclusion criteria required participants to be older than 18 years. Tourists or students from other nations were excluded. Additionally, 16 participants were excluded due to missing self-reported anthropometric data on body weight and height. The final sample comprised 9,036 European participants.

After calculating BMI, participants were divided into two groups: those with a BMI < 25.0 kg/m$^2$ (N = 6,015) and those with a BMI ≥ 25.0 kg/m$^2$ (N = 3,021).

Before recruitment, participants were informed of the study's goals and assured that participation was voluntary, and that their responses would remain anonymous and confidential. After being provided with all the information needed to make an informed decision, participants were given a reasonable amount of time to consider participating in the study. During this period, the researchers were available to answer any questions related to the study. Participants who agreed to participate provided verbal informed consent. For each participant agreeing to participate, the researcher who conducted the informed consent procedure filled out the oral consent template and signed the document in the participant's presence to confirm their consent. Afterward, participants completed the survey in private cubicles to maintain their privacy. Data collection took place from 16 October 2017 to 9 March 2018.

## Inclusivity in global research

Ethical approval was granted by the Ethical Committee of the Polytechnic Institute of Viseu (registration number 04/2017), the project holder. The study adhered to the ethical guidelines of the Declaration of Helsinki and was approved for ethical purposes and for application in the following institutions of each participating country, in addition to that of the project holder: General Hospital Zadar, Croatia; Alexander Technological Educational Institute of Thessaloniki (ATEITh), School of Agriculture Technology, Food Technology and Nutrition, Greece; Hungarian Chamber of Agriculture, Hungary; University of Parma, Italy; Latvia University of Agriculture, Faculty of Food Technology, Department of Food Technology, Latvia; The Netherlands Food Quality and Design Group, Wageningen University & Research, The Netherlands; Wroclaw University of Environmental and Life Sciences, Faculty of Food Science, Poland; University of Medicine and Pharmacy Tirgu-Mures, Faculty of Medicine, Romania; University of Belgrade, Faculty of Agriculture, Republic of Serbia; University of Primorska, Faculty of Health Sciences, Department of Nutritional Counselling-Dietetics, Slovenia. Approval from all Ethics Committees was obtained before data collection began.

Additional information regarding the ethical, cultural, and scientific considerations specific to inclusivity in global research is included in the Supporting Information (S1 Table).

## Questionnaire

The emotional components of food consumption in 12 different European nations were assessed using the paper-and-pencil "The Motivations for Food Choices" (EATMOT) questionnaire.

The final version of the working instrument was developed, tested, and validated on a sample from the Portuguese population [21]. The working instrument was then provided to all other nations in English, which they subsequently translated into their native languages without changing the questions, using alternatives, or altering the number of questions. For better accuracy, a back-translation method was employed: experts first translated the questionnaire into the target

language, after which other individuals translated it back into English. The final questionnaire was then sent to the project manager in Portugal for approval and verification to ensure no changes had been made to the initial format. Therefore, the results and responses were comparable across all nations. The validation of the questionnaire has been described in our previous papers [22,23].

Before distributing the questionnaire, a small pilot test was conducted in each of the participating nations, involving ten participants from various age groups, to ensure clarity and confirm that the findings fell within the expected range. The questionnaire took an average of 15 minutes to complete.

The questionnaire included sociodemographic data, self-assessment of health, MHB data, and EMFC data.

Sociodemographic data included country of residence, place of residence, age, gender, education level, marital status, employment status, and professional occupation.

Health assessment involved self-assessment of health, including anthropometric data, physical activity, time spent daily in front of a computer or watching television, adherence to a balanced/healthy diet, and the presence of chronic diseases (CVDs, DM, hypercholesterolemia, hypertension, GDs, IDs, obesity, and other diseases). Self-assessed anthropometric data included body weight in kilograms and body height in meters. BMI was calculated using the formula: BMI = body weight (kg)/ body height$^2$ (m²).

The Likert scale was used to categorize MHB and EMFC-related issues, ranked from 1 to 5: 1 - strongly disagree, 2 - disagree, 3 - neither agree nor disagree, 4 - agree, and 5 - strongly agree.

MHB included 10 items on food hygiene and safety, a healthy and balanced diet, low-fat diet, low-cholesterol diet, low-sugar diet, additive-free foods, processed foods, genetically modified organism-free foods, a vitamin- and mineral-rich diet, and foods that promote health. MHB was calculated as the sum of all 10 responses (range 10–50); with a higher score indicating a stronger level of MHB.

EMFC contained information on the emotional impact of food consumption, eating to cope with stress, depression, boredom, loneliness, emotional consolation, and the perception that food helps control body weight, stay awake and alert, contributes to relaxation, and creates a good feeling. Overall, EMFC was the sum of all 9 responses (range 9–45); with a higher value indicating a higher frequency of food consumption in response to certain emotional states.

## Statistical analysis

Statistical analysis was conducted using SPSS Statistics v21.0 (IBM, Armonk, NY, USA) on a final sample of 9,036 European participants. A p-value of < 0.05 was considered statistically significant.

The Kolmogorov–Smirnov test was used to assess the data distribution. For numerical variables, the median and inter-quartile range were computed, depending on the distribution. Categorical variables were presented as absolute numbers and percentages. The Chi-square test was used for categorical variables, and the Mann-Whitney U test was used for numerical variables to investigate differences between groups.

The multivariate analysis included both linear and logistic regression to examine associations between variables. Three separate multiple linear regression models were applied to identify factors associated with the linear dependent variables: BMI sum, MHB, and EMFC. BMI was included as a continuous variable in all linear regression models. Predictors in these three models included: place of residence [12 European countries, with Portugal as the reference group (RG) due to the highest incidence of normal BMI and MHB], age, gender (with "male" as the RG), education level (with "no university education" as the RG), marital status (with "married" as the RG), employment status (with "employed" as the RG), professional occupation (with "other professions" as the RG), responsibility for buying food (with "no" as the RG), physical activity and sitting in front of a TV or computer (with "no" as the RG), following a balanced/healthy diet (with "never/rarely" as the RG), emotional reasons for eating to cope with stress, depression, boredom, loneliness, emotional consolation, help control weight, stay awake and alert, relax, and feel good (with "no" as the RG), and the presence of CVDs, DM, hypercholesterolemia, hypertension, GDs, IDs, obesity, and other diseases (with "no" as the RG).

The effect size was calculated to measure the strength of the association between the two variables: MHB and EMFC, BMI and EMFC, and BMI and MHB.

Several logistic regression models were used to identify characteristics associated with binary outcome variables: BMI, CVDs, DM, hypercholesterolemia, hypertension, GDs, IDs, and obesity (each in a separate logistic regression model). BMI was dichotomized according to the World Health Organization (WHO) classification, as described in the "Study design and participants" section, to ensure adherence to established public health standards and to facilitate the interpretation of results. However, it is important to note that a BMI of $25.0\,kg/m^2$ or higher encompasses both overweight and obesity, with obesity defined as a BMI of $30.0\,kg/m^2$ or greater. Conversely, a BMI below $25.0\,kg/m^2$ does not necessarily indicate normal body weight. For instance, a BMI of $< 18.5\,kg/m^2$ reflects malnutrition rather than a healthy weight. This highlights the need to interpret BMI values within the broader context of individual health and nutritional status. NCD variables were also dichotomized (presence or absence of disease) to ensure interpretability within the logistic regression model. Predictors in all these models included the same variables as those in the linear regression models mentioned above. RGs were also defined as in the previous regression models. Beta values were converted into odds ratios with 95% confidence intervals (lower and upper bounds).

To minimize statistical bias, all predictors were included simultaneously in the regression models. In certain models, the primary outcome variables were also incorporated as predictors in the analysis.

## Results

### Sociodemographic characteristics, body mass index, and lifestyles of the European population

The sociodemographic characteristics of the entire European sample are detailed in S2 Table. All European countries had a normal median BMI, except Hungary [median (Mdn) = $25.9\,kg/m^2$; interquartile range (IQR) = 7.1], which exhibited the highest percentage of participants with a BMI $\geq 30.0\,kg/m^2$ (24.0%). Participants from Portugal had the highest proportion of individuals with a normal BMI (79.9%) compared to other countries, while 7.2% of Serbian adults had a BMI below $18.5\,kg/m^2$. Lithuania showed the highest prevalence of food consumption under negative emotional conditions, while Hungary had the lowest. Lithuanians exhibited the most frequent food consumption in response to stress, depression, loneliness, boredom, and the need for comfort, while Portuguese participants demonstrated the highest MHB (Mdn = 39.0; IQR = 7.0) compared to other European countries. On the other hand, the lowest EMFC was recorded in Hungary (Mdn = 23.0; IQR = 10.0), Serbia (Mdn = 24.0; IQR = 9.0), and Portugal (Mdn = 24.0; IQR = 6.0) (Table 1). The classification of BMI, lifestyle patterns, MHB, and EMFC for the entire sample is described in Table 1.

### Association between body mass index, motivation for health behavior, and emotional motivation for food consumption

The linear regression model confirmed associations between BMI, MHB, and EMFC. Compared to Portugal, all other European countries exhibited higher BMI, particularly Slovenia (ß = 0.15; t = 12.95; p < 0.001), Latvia (ß = 0.12; t = 11.45; p < 0.001), and Croatia (ß = 0.12; t = 9.74; p < 0.001). Most European countries showed lower MHB, except for Lithuania (ß = 0.06; t = 5.74; p < 0.001) and Romania (ß = 0.04; t = 3.18; p = 0.001), which had slightly higher MHB compared to Portugal. Compared to Portugal, EMFC was higher in Croatia, Greece, Italy, Latvia, Lithuania, Poland, and Romania, but lower in Serbia. Hungary, the Netherlands, and Slovenia showed no significant difference in EMFC compared to Portugal (Table 2).

Age was positively associated with BMI (ß = 0.28; t = 25.53; p < 0.001) and MHB (ß = 0.17; t = 14.92; p < 0.001), while its negative association with EMFC was borderline insignificant (ß = -0.01; t = -1.68; p = 0.093). Female gender, living in an urban environment, university education, unemployment, and being single, widowed, or divorced were associated with lower BMI. Additionally, female gender contributed to higher MHB. Female gender (ß = 0.02; t = 3.78; p < 0.001),

**Table 1. Body mass index, lifestyle patterns, and motivation for food consumption in the European population (N=9,036).**

| | | Croatia (N=1,538) | Greece (N=498) | Hungary (N=500) | Italy (N=541) | Latvia (N=634) | Lithuania (N=507) | Netherlands (N=521) | Poland (N=583) | Portugal (N=1,304) | Romania (N=821) | Serbia (N=498) | Slovenia (N=1,091) | Overall p |
|---|---|---|---|---|---|---|---|---|---|---|---|---|---|---|
| BMI, Mdn (IQR) | | 23.5 (5.4) | 22.9 (4.7) | 25.9 (7.1) | 23.1 (5.0) | 24.3 (5.3) | 23.5 (6.1) | 23.3 (5.3) | 23.1 (5.2) | 21.5 (3.2) | 24.2 (5.6) | 22.3 (4.7) | 23.6 (5.3) | < 0.001* |
| BMI classification, N (%) | Underweight (< 18.5 kg/m²) | 52 (3.4) | 14 (2.8) | 13 (2.6) | 32 (5.9) | 15 (2.4) | 27 (5.3) | 19 (3.6) | 26 (4.5) | 72 (5.5) | 42 (5.1) | 36 (7.2) | 40 (3.7) | < 0.001† |
| | Normal weight (18.5–24.9 kg/m²) | 922 (59.9) | 340 (68.3) | 201 (40.2) | 342 (63.2) | 354 (55.8) | 285 (56.2) | 325 (62.4) | 371 (63.6) | 1042 (79.9) | 442 (53.8) | 344 (69.1) | 659 (60.4) | |
| | Overweight (25.0–29.9 kg/m²) | 463 (30.1) | 117 (23.5) | 166 (33.2) | 130 (24.0) | 176 (27.8) | 144 (28.4) | 113 (21.7) | 133 (22.8) | 148 (11.3) | 242 (29.5) | 96 (19.3) | 270 (24.7) | |
| | Obesity (≥ 30.0 kg/m²) | 101 (6.6) | 27 (5.4) | 120 (24.0) | 37 (6.8) | 89 (14.0) | 51 (10.1) | 64 (12.3) | 53 (9.1) | 42 (3.2) | 95 (11.6) | 22 (4.4) | 122 (11.2) | |
| Following a balanced diet, Mdn (IQR) | | 3.0 (1.0) | 4.0 (1.0) | 3.0 (1.0) | 4.0 (1.0) | 3.0 (1.0) | 3.0 (1.0) | 4.0 (1.0) | 3.0 (1.0) | 3.0 (1.0) | 3.0 (1.0) | 3.0 (1.0) | 4.0 (0.0) | < 0.001* |
| Sitting in front of computer or TV (hours/day); Mdn (IQR) | | 3.0 (4.0) | 3.0 (3.0) | 3.0 (3.0) | 4.0 (6.0) | 6.0 (5.0) | 3.0 (3.0) | 4.0 (3.3) | 4.0 (6.0) | 6.0 (5.0) | 4.0 (5.0) | 3.0 (3.0) | 2.0 (3.0) | < 0.001* |
| Exercise, N (%) | No | 598 (38.9) | 176 (35.3) | 263 (52.6) | 221 (40.9) | 201 (31.7) | 286 (56.4) | 177 (34.0) | 214 (36.7) | 295 (22.6) | 324 (39.5) | 167 (33.5) | 198 (18.1) | < 0.001† |
| | Yes | 940 (61.1) | 322 (64.7) | 237 (47.4) | 320 (59.1) | 433 (68.3) | 221 (43.6) | 344 (66.0) | 369 (63.3) | 1009 (77.4) | 497 (60.5) | 331 (66.5) | 893 (81.9) | |
| Eating when stressed, N (%) | No | 1098 (71.4) | 328 (65.9) | 420 (84.0) | 343 (63.4) | 425 (67.0) | 277 (54.6) | 333 (63.9) | 385 (66.0) | 1004 (77.0) | 535 (65.2) | 334 (67.1) | 677 (62.1) | < 0.001† |
| | Yes | 440 (28.6) | 170 (34.1) | 80 (16.0) | 198 (36.6) | 209 (33.0) | 230 (45.4) | 188 (36.1) | 198 (34.0) | 300 (23.0) | 286 (34.8) | 164 (32.9) | 414 (37.9) | |
| Eating when depressed, N (%) | No | 1075 (69.9) | 315 (63.3) | 368 (73.6) | 317 (58.6) | 450 (71.0) | 289 (57.0) | 320 (61.4) | 379 (65.0) | 1014 (77.8) | 513 (62.5) | 355 (71.3) | 653 (59.9) | < 0.001† |
| | Yes | 463 (30.1) | 183 (36.7) | 132 (26.4) | 224 (41.4) | 184 (29.0) | 218 (43.0) | 201 (38.6) | 204 (35.0) | 290 (22.2) | 308 (37.5) | 143 (28.7) | 438 (40.1) | |
| Eating out of boredom, N (%) | No | 885 (57.5) | 265 (53.2) | 379 (75.8) | 273 (50.5) | 327 (51.6) | 176 (34.7) | 208 (39.9) | 273 (46.8) | 920 (70.6) | 455 (55.4) | 266 (53.4) | 629 (57.7) | < 0.001† |
| | Yes | 653 (42.5) | 233 (46.8) | 121 (24.2) | 268 (49.5) | 307 (48.4) | 331 (65.3) | 313 (60.1) | 310 (53.2) | 384 (29.4) | 366 (44.6) | 232 (46.6) | 462 (42.3) | |
| Eating when lonely, N (%) | No | 1326 (86.2) | 396 (79.5) | 438 (87.6) | 434 (80.2) | 496 (78.2) | 319 (62.9) | 388 (74.5) | 479 (82.2) | 1156 (88.7) | 606 (73.8) | 416 (83.5) | 890 (81.6) | < 0.001† |
| | Yes | 212 (13.8) | 102 (20.5) | 62 (12.4) | 107 (19.8) | 138 (21.8) | 188 (37.1) | 133 (25.5) | 104 (17.8) | 148 (11.3) | 215 (26.2) | 82 (16.5) | 201 (18.4) | |
| Food serves as an emotional consolation, N (%) | No | 1171 (76.1) | 391 (78.5) | 427 (85.4) | 413 (76.3) | 453 (71.5) | 316 (62.3) | 399 (76.6) | 436 (74.8) | 1095 (84.0) | 625 (76.1) | 431 (86.5) | 853 (78.2) | < 0.001† |
| | Yes | 367 (23.9) | 107 (21.5) | 73 (14.6) | 128 (23.7) | 181 (28.5) | 191 (37.7) | 122 (23.4) | 147 (25.2) | 209 (16.0) | 196 (23.9) | 67 (13.5) | 238 (21.8) | |

*(Continued)*

**Table 1.** (Continued)

| | | Croatia (N=1,538) | Greece (N=498) | Hungary (N=500) | Italy (N=541) | Latvia (N=634) | Lithuania (N=507) | Netherlands (N=521) | Poland (N=583) | Portugal (N=1,304) | Romania (N=821) | Serbia (N=498) | Slovenia (N=1,091) | Overall p |
|---|---|---|---|---|---|---|---|---|---|---|---|---|---|---|
| Food helps control weight, N (%) | No | 960 (62.4) | 318 (63.9) | 372 (74.4) | 342 (63.2) | 450 (71.0) | 340 (67.1) | 368 (70.6) | 301 (51.6) | 403 (30.9) | 428 (52.1) | 365 (73.3) | 520 (47.7) | < 0.001† |
| | Yes | 578 (37.6) | 180 (36.1) | 128 (25.6) | 199 (36.8) | 184 (29.0) | 167 (32.9) | 153 (29.4) | 282 (48.4) | 901 (69.1) | 393 (47.9) | 133 (26.7) | 571 (52.3) | |
| Food keeps one awake and alert, N (%) | No | 1025 (66.6) | 351 (70.5) | 311 (62.2) | 378 (69.9) | 392 (61.8) | 287 (56.6) | 390 (74.9) | 395 (67.8) | 1043 (80.0) | 450 (54.8) | 279 (56.0) | 790 (72.4) | < 0.001† |
| | Yes | 513 (33.4) | 147 (29.5) | 189 (37.8) | 163 (30.1) | 242 (38.2) | 220 (43.4) | 131 (25.1) | 188 (32.2) | 261 (20.0) | 371 (45.2) | 219 (44.0) | 301 (27.6) | |
| Food helps with relaxation, N (%) | No | 995 (64.7) | 295 (59.2) | 390 (78.0) | 324 (59.9) | 384 (60.6) | 258 (50.9) | 380 (72.9) | 297 (50.9) | 476 (36.5) | 539 (65.7) | 254 (51.0) | 774 (70.9) | < 0.001† |
| | Yes | 543 (35.3) | 203 (40.8) | 110 (22.0) | 217 (40.1) | 250 (39.4) | 249 (49.1) | 141 (27.1) | 286 (49.1) | 828 (63.5) | 282 (34.3) | 244 (49.0) | 317 (29.1) | |
| Food makes one feel good, N (%) | No | 592 (38.5) | 168 (33.7) | 269 (53.8) | 159 (29.4) | 160 (25.2) | 116 (22.9) | 190 (36.5) | 176 (30.2) | 270 (20.7) | 375 (45.7) | 238 (47.8) | 465 (42.6) | < 0.001† |
| | Yes | 946 (61.5) | 330 (66.3) | 231 (46.2) | 382 (70.6) | 474 (74.8) | 391 (77.1) | 331 (63.5) | 407 (69.8) | 1034 (79.3) | 446 (54.3) | 260 (52.2) | 626 (57.4) | |
| Emotional motivation for food consumption, Mdn (IQR) | | 25.0 (9.0) | 27.0 (9.0) | 23.0 (10.0) | 27.0 (8.0) | 28.5 (9.0) | 30.0 (8.0) | 26.0 (8.0) | 27.0 (7.0) | 24.0 (6.0) | 27.0 (10.0) | 24.0 (9.0) | 26.0 (9.0) | < 0.001* |
| Motivation for health behavior, Mdn (IQR) | | 34.0 (7.0) | 34.0 (8.0) | 31.0 (7.0) | 34.0 (6.0) | 34.0 (7.0) | 36.0 (7.0) | 30.0 (7.0) | 35.0 (6.0) | 39.0 (7.0) | 30.0 (8.0) | 33.0 (7.0) | 35.0 (6.0) | < 0.001* |

Note: BMI=Body Mass Index; Mdn (IQR) = Median (Interquartile Range); N (%) = absolute number (percentage); Motivation for food consumption: range=9–45; *Kruskal Wallis test; †Chi Square test.

**Table 2.** Associations between body mass index, motivation for health behavior, and emotional motivation for food consumption in linear regression models.

| | BMI | | | MHB | | | EMFC | | |
|---|---|---|---|---|---|---|---|---|---|
| | Beta | t | p | Beta | t | p | Beta | t | p |
| Country (Portugal was the referent group) | | | | | | | | | |
| Croatia | 0.12 | 9.74 | < 0.001 | -0.09 | -7.16 | < 0.001 | 0.02 | 2.73 | 0.006 |
| Greece | 0.08 | 8.03 | < 0.001 | -0.07 | -7.07 | < 0.001 | 0.02 | 3.98 | < 0.001 |
| Hungary | 0.08 | 7.65 | < 0.001 | -0.16 | -15.02 | < 0.001 | 0.01 | 1.04 | 0.300 |
| Italy | 0.02 | 2.19 | 0.029 | -0.08 | -7.74 | < 0.001 | 0.03 | 4.34 | < 0.001 |
| Latvia | 0.12 | 11.45 | < 0.001 | -0.09 | -8.59 | < 0.001 | 0.10 | 16.44 | < 0.001 |
| Lithuania | 0.07 | 7.01 | < 0.001 | 0.06 | 5.74 | < 0.001 | 0.03 | 4.62 | < 0.001 |
| Netherlands | 0.09 | 9.11 | < 0.001 | -0.20 | -19.70 | < 0.001 | 0.00 | -0.68 | 0.499 |
| Poland | 0.05 | 5.42 | < 0.001 | -0.05 | -5.23 | < 0.001 | 0.02 | 3.15 | 0.002 |
| Romania | 0.09 | 7.96 | < 0.001 | 0.04 | 3.18 | 0.001 | 0.03 | 4.12 | < 0.001 |
| Serbia | 0.05 | 4.80 | < 0.001 | -0.07 | -6.59 | < 0.001 | -0.05 | -7.55 | < 0.001 |
| Slovenia | 0.15 | 12.95 | < 0.001 | -0.10 | -8.62 | < 0.001 | 0.00 | 0.40 | 0.686 |
| Age | 0.28 | 25.53 | < 0.001 | 0.17 | 14.92 | < 0.001 | -0.01 | -1.68 | 0.093 |
| Gender (male was the referent group) | | | | | | | | | |
| Female | -0.20 | -21.80 | < 0.001 | 0.07 | 6.97 | < 0.001 | 0.02 | 3.78 | < 0.001 |
| Environment (rural was the referent group) | | | | | | | | | |
| Urban | -0.06 | -6.71 | < 0.001 | 0.00 | -0.31 | 0.755 | 0.00 | 0.43 | 0.670 |
| Education level (no university was the referent group) | | | | | | | | | |
| University | -0.06 | -6.44 | < 0.001 | 0.00 | 0.10 | 0.921 | 0.01 | 1.71 | 0.086 |
| Employment (employed was the referent group) | | | | | | | | | |
| Unemployed | -0.03 | -3.28 | 0.001 | -0.02 | -1.80 | 0.072 | 0.02 | 3.08 | 0.002 |
| Marital status (married was the referent group) | | | | | | | | | |
| Single, divorced, widowed | -0.03 | -3.23 | 0.001 | -0.02 | -1.70 | 0.089 | 0.01 | 2.36 | 0.018 |
| Professional occupation (other was the referent group) | | | | | | | | | |
| Nutrition and food | 0.01 | 1.60 | 0.110 | -0.04 | -3.83 | < 0.001 | 0.00 | -0.83 | 0.407 |
| Agriculture | 0.03 | 3.59 | < 0.001 | -0.01 | -1.54 | 0.124 | -0.01 | -1.25 | 0.210 |
| Sport | 0.00 | -0.57 | 0.569 | -0.01 | -1.69 | 0.091 | 0.01 | 2.58 | 0.010 |
| Psychology | 0.00 | 0.48 | 0.634 | 0.00 | -0.01 | 0.995 | 0.00 | -0.32 | 0.752 |
| Health | 0.00 | -0.54 | 0.591 | -0.02 | -2.35 | 0.019 | -0.01 | -1.45 | 0.148 |
| Responsible for buying food (no was the referent group) | | | | | | | | | |
| Yes | -0.02 | -1.89 | 0.058 | -0.03 | -3.49 | < 0.001 | 0.02 | 3.57 | < 0.001 |
| Exercise (no was the referent group) | | | | | | | | | |
| Yes | -0.05 | -5.37 | < 0.001 | 0.08 | 8.61 | < 0.001 | 0.00 | 0.63 | 0.526 |
| TV or computer, (hours/day) | 0.00 | 0.13 | 0.897 | -0.04 | -4.88 | < 0.001 | 0.00 | 0.77 | 0.443 |
| Balanced/healthy diet (never/rarely was the referent group) | | | | | | | | | |
| Often/always | 0.00 | 0.20 | 0.841 | 0.21 | 23.66 | < 0.001 | 0.00 | -0.91 | 0.362 |
| EMFC | | | | | | | | | |
| Cope with stress | 0.04 | 3.57 | < 0.001 | -0.02 | -2.10 | 0.036 | 0.20 | 32.49 | < 0.001 |
| Depression | -0.02 | -1.83 | 0.067 | 0.01 | 1.20 | 0.229 | 0.21 | 34.77 | < 0.001 |
| Boredom | 0.01 | 0.72 | 0.471 | -0.05 | -4.76 | < 0.001 | 0.23 | 39.53 | < 0.001 |
| Loneliness | 0.03 | 2.22 | 0.026 | 0.00 | -0.18 | 0.858 | 0.15 | 22.15 | < 0.001 |
| Emotional consolation | 0.03 | 2.64 | 0.008 | -0.02 | -1.29 | 0.198 | 0.14 | 21.07 | < 0.001 |
| Helps control weight | -0.04 | -3.80 | < 0.001 | 0.28 | 30.22 | < 0.001 | 0.12 | 21.86 | < 0.001 |
| Keeps awake and alert | -0.01 | -0.61 | 0.540 | -0.08 | -7.87 | < 0.001 | 0.18 | 34.17 | < 0.001 |

*(Continued)*

**Table 2.** (Continued)

| | BMI | | | MHB | | | EMFC | | |
|---|---|---|---|---|---|---|---|---|---|
| | Beta | t | p | Beta | t | p | Beta | t | p |
| Relaxation | -0.05 | -5.27 | < 0.001 | 0.05 | 5.29 | < 0.001 | 0.17 | 32.11 | < 0.001 |
| Makes one feel good | -0.05 | -5.76 | < 0.001 | 0.05 | 5.16 | < 0.001 | 0.14 | 25.36 | < 0.001 |
| Chronic noncommunicable diseases | | | | | | | | | |
| Cardiovascular diseases | -0.01 | -0.71 | 0.477 | -0.01 | -1.09 | 0.275 | -0.01 | -1.45 | 0.146 |
| Diabetes mellitus | 0.04 | 5.22 | < 0.001 | 0.00 | 0.44 | 0.660 | 0.00 | 0.38 | 0.707 |
| Hypercholesterolemia | -0.03 | -3.84 | < 0.001 | 0.00 | -0.50 | 0.616 | 0.00 | -0.55 | 0.579 |
| Hypertension | 0.04 | 4.51 | < 0.001 | 0.02 | 2.29 | 0.022 | 0.00 | 0.12 | 0.903 |
| Gastric disorders | 0.00 | -0.31 | 0.760 | -0.01 | -1.50 | 0.133 | 0.00 | 0.22 | 0.829 |
| Intestinal disorders | -0.01 | -1.63 | 0.102 | 0.02 | 2.34 | 0.019 | 0.00 | -0.08 | 0.933 |
| Obesity | 0.35 | 40.48 | < 0.001 | -0.01 | -0.87 | 0.385 | 0.01 | 1.15 | 0.248 |
| Other diseases | 0.04 | 4.88 | < 0.001 | 0.00 | 0.18 | 0.861 | 0.01 | 1.44 | 0.151 |
| BMI | – | – | – | -0.05 | -4.38 | < 0.001 | 0.05 | 7.70 | < 0.001 |
| MHB | -0.05 | -4.38 | < 0.001 | – | – | – | 0.00 | -0.21 | 0.833 |
| EMFC | 0.13 | 7.70 | < 0.001 | 0.00 | -0.21 | 0.833 | – | – | – |
| Overall model significance | R=0.63; R²=0.39; p<0.001 | | | R=0.61; R²=0.37; p<0.001 | | | R=0.88; R²=0.78; p<0.001 | | |
| Effect Size Cohen's f² | 0.80 | | | 0.77 | | | 1.88 | | |

Note: BMI = Body Mass Index; MHB = Motivation for health behavior; EMFC = Emotional motivation for food consumption.

unemployment (ß=0.02; t=3.08; p=0.002), and being single, widowed, or divorced (ß=0.01; t=2.36; p=0.018) were associated with higher EMFC.

Exercise was negatively associated with BMI (ß=-0.05; t=-5.37; p<0.001) and positively associated with MHB (ß=0.08; t=8.61; p<0.001). Consumption of a balanced and healthy diet was positively associated with MHB (ß=0.21; t=23.66; p<0.001).

Using food as a coping mechanism for stress (ß=0.04; t=3.57; p<0.001), loneliness (ß=0.03; t=2.22; p=0.026), and emotional consolation (ß=0.03; t=2.64; p=0.008) was positively associated with BMI. Conversely, eating food to help control weight (ß=-0.04; t=-3.80; p<0.001), promote relaxation (ß=-0.05; t=-5.27; p<0.001), and induce positive feelings (ß=-0.05; t=-5.76; p<0.001) was negatively associated with BMI.

Consuming food that helps control weight (ß=0.28; t=30.22; p<0.001), promotes relaxation (ß=0.05; t=5.29; p<0.001), and induces positive feelings (ß=0.05; t=5.16; p<0.001) positively contributed to MHB. In contrast, using food as a coping mechanism for stress, boredom, and staying awake and alert was negatively associated with MHB.

DM (ß=0.04; t=5.22; p<0.001), hypertension (ß=0.04; t=4.51; p<0.001), obesity (ß=0.35; t=40.48; p<0.001), and other non-specific diseases (ß=0.04; t=4.88; p<0.001) contributed to higher BMI. Additionally, hypertension (ß=0.02; t=2.29; p=0.022) and IDs (ß=0.02; t=2.34; p=0.019) were associated with higher MHB. No statistically significant associations were observed between the incidence of NCDs and EMFC.

No associations were found between MHB and EMFC. However, BMI was negatively associated with MHB (ß=-0.05; t=-4.38; p<0.001) and positively associated with EMFC (ß=0.13; t=7.70; p<0.001) (Table 2).

## Association between body mass index and chronic noncommunicable diseases

When predictors were considered in the multivariate logistic regression model, several characteristics were associated with a higher BMI and the incidence of NCDs (Table 3).

**Table 3. Logistic regression model of characteristics associated with body mass index and chronic noncommunicable diseases (*OR (95% CI); p), N=9,036.**

| | BMI ≥ 25.0 kg/m² | Cardiovascular diseases | Diabetes mellitus | Hypercholesterolemia | Hypertension | Gastric disorders | Intestinal disorders | Obesity |
|---|---|---|---|---|---|---|---|---|
| Country (Portugal was the referent group) | | | | | | | | |
| Croatia | 3.48 (2.78-4.34); <0.001 | 2.68 (1.56-4.60); <0.001 | 0.86 (0.46-1.59); 0.625 | 0.33 (0.23-0.48); <0.001 | 0.19 (0.14-0.28); <0.001 | 1.90 (1.14-3.17); 0.014 | 1.50 (0.80-2.82); 0.206 | 0.97 (0.53-1.80); 0.933 |
| Greece | 3.31 (2.47-4.45); <0.001 | 0.18 (0.02-1.39); 0.101 | 0.00 (0.00-0.00); 0.992 | 0.29 (0.15-0.55); <0.001 | 0.17 (0.09-0.33); <0.001 | 1.27 (0.61-2.66); 0.527 | 2.85 (1.37-5.92); 0.005 | 1.15 (0.51-2.61); 0.741 |
| Hungary | 3.55 (2.67-4.72); <0.001 | 2.23 (1.18-4.21); 0.014 | 2.31 (1.23-4.35); 0.010 | 0.36 (0.22-0.57); <0.001 | 1.25 (0.86-1.81); 0.234 | 1.12 (0.57-2.20); 0.749 | 4.12 (2.06-8.27); <0.001 | 4.15 (2.26-7.62); <0.001 |
| Italy | 1.62 (1.23-2.15); 0.001 | 0.57 (0.25-1.30); 0.180 | 0.53 (0.22-1.26); 0.149 | 0.44 (0.28-0.68); <0.001 | 0.19 (0.12-0.31); <0.001 | 2.12 (1.17-3.84); 0.013 | 2.87 (1.47-5.63); 0.002 | 0.87 (0.40-1.88); 0.719 |
| Latvia | 4.20 (3.21-5.50); <0.001 | 2.63 (1.33-5.20); 0.005 | 0.63 (0.26-1.52); 0.304 | 0.23(0.13-0.41); <0.001 | 0.28 (0.17-0.44); <0.001 | 2.27 (1.25-4.13); 0.007 | 3.24 (1.63-6.46); 0.001 | 1.14 (0.58-2.22); 0.712 |
| Lithuania | 3.76 (2.82-5.01); <0.001 | 2.39 (1.21-4.70); 0.012 | 0.37 (0.14-0.99); 0.048 | 0.14 (0.07-0.27); <0.001 | 0.33 (0.20-0.55); <0.001 | 2.54 (1.38-4.70); 0.003 | 0.30 (0.07-1.35); 0.117 | 1.29 (0.63-2.62); 0.484 |
| Netherlands | 3.45 (2.58-4.62); <0.001 | 1.73 (0.80-3.72); 0.161 | 1.85 (0.91-3.79); 0.091 | 0.13 (0.06-0.26); <0.001 | 0.39 (0.24-0.63); <0.001 | 0.88 (0.42-1.87); 0.748 | 3.38 (1.65-6.96); 0.001 | 1.20 (0.58-2.46); 0.619 |
| Poland | 2.60 (1.97-3.45); <0.001 | 0.43 (0.14-1.29); 0.131 | 1.24 (0.58-2.63); 0.576 | 0.35 (0.21-0.59); <0.001 | 0.28 (0.17-0.46); <0.001 | 1.33 (0.69-2.59); 0.397 | 3.99 (2.09-7.59); <0.001 | 3.33 (1.75-6.33); <0.001 |
| Romania | 3.39 (2.63-4.37); <0.001 | 2.14 (1.15-3.97); 0.016 | 1.35 (0.70-2.58); 0.367 | 0.53 (0.35-0.79); 0.002 | 0.24 (0.16-0.36); <0.001 | 2.97 (1.74-5.05); <0.001 | 2.62 (1.37-5.01); 0.004 | 2.77 (1.50-5.12); 0.001 |
| Serbia | 2.65 (1.93-3.65); <0.001 | 1.76 (0.79-3.96); 0.168 | 1.11 (0.48-2.59); 0.804 | 0.32 (0.17-0.60); <0.001 | 0.38 (0.21-0.67); 0.001 | 1.39 (0.68-2.81); 0.364 | 0.90 (0.34-2.42); 0.838 | 5.11 (2.55-10.26); <0.001 |
| Slovenia | 4.30 (3.38-5.48); <0.001 | 1.29 (0.61-2.70); 0.504 | 0.85 (0.41-1.75); 0.653 | 0.34 (0.21-0.54); <0.001 | 0.24 (0.16-0.38); <0.001 | 1.08 (0.59-1.96); 0.811 | 2.42 (1.27-4.58); 0.007 | 3.40 (1.87-6.18); <0.001 |
| Age | 1.05 (1.04-1.05); <0.001 | 1.07 (1.06-1.08); <0.001 | 1.04 (1.02-1.05); <0.001 | 1.06 (1.06-1.07); <0.001 | 1.08 (1.07-1.09); <0.001 | 1.01(1.00-1.02); 0.150 | 1.01 (1.00-1.02); 0.038 | 1.01 (1.00-1.02); 0.006 |
| Gender (male was the referent group) | | | | | | | | |
| Female | 0.33 (0.29-0.37); <0.001 | 0.69 (0.51-0.93); 0.015 | 0.72 (0.52-1.00); 0.051 | 0.69 (0.55-0.86); 0.001 | 1.10 (0.89-1.36); 0.395 | 1.66 (1.23-2.25); 0.001 | 1.36 (0.96-1.92); 0.081 | 1.48 (1.14-1.92); 0.003 |
| Environment (rural was the referent group) | | | | | | | | |
| Urban | 0.72 (0.62-0.84); <0.001 | 0.84 (0.58-1.21); 0.346 | 1.39 (0.91-2.15); 0.131 | 1.01 (0.74-1.38); 0.939 | 1.17 (0.88-1.55); 0.284 | 0.95 (0.69-1.31); 0.757 | 1.20 (0.81-1.78); 0.361 | 0.78 (0.59-1.03); 0.084 |
| Education level (no university was the referent group) | | | | | | | | |
| University | 0.76 (0.68-0.86); <0.001 | 0.90 (0.67-1.22); 0.514 | 0.86 (0.61-1.20); 0.380 | 1.27 (1.00-1.61); 0.051 | 1.03 (0.83-1.27); 0.777 | 0.93 (0.72-1.22); 0.610 | 0.96 (0.71-1.31); 0.805 | 0.99 (0.77-1.27); 0.946 |
| Employment (employed was the referent group) | | | | | | | | |
| Unemployed | 0.84 (0.74-0.96); 0.009 | 1.53 (1.13-2.08); 0.006 | 1.18 (0.83-1.66); 0.356 | 1.11 (0.85-1.43); 0.446 | 0.86 (0.68-1.09); 0.212 | 1.24 (0.94-1.64); 0.120 | 1.03 (0.75-1.43); 0.839 | 0.85 (0.65-1.12); 0.254 |

*(Continued)*

**Table 3.** (Continued)

| | BMI ≥ 25.0 kg/m² | Cardiovascular diseases | Diabetes mellitus | Hypercholesterolemia | Hypertension | Gastric disorders | Intestinal disorders | Obesity |
|---|---|---|---|---|---|---|---|---|
| Marital status (married was the referent group) | | | | | | | | |
| Single, widowed, divorced | 0.80 (0.71-0.90); <0.001 | 1.25 (0.94-1.67); 0.129 | 0.79 (0.57-1.10); 0.166 | 0.84 (0.67-1.06); 0.138 | 1.02 (0.83-1.25); 0.851 | 0.90 (0.69-1.16); 0.411 | 1.07 (0.80-1.43); 0.651 | 1.06 (0.83-1.35); 0.655 |
| Professional occupation (other was the referent group) | | | | | | | | |
| Nutrition and food | 0.98 (0.85-1.14); 0.829 | 1.08 (0.71-1.65); 0.708 | 0.84 (0.52-1.34); 0.462 | 1.04 (0.75-1.44); 0.804 | 0.71 (0.52-0.98); 0.035 | 1.06 (0.77-1.45); 0.729 | 0.89 (0.62-1.27); 0.526 | 1.19 (0.87-1.61); 0.275 |
| Agriculture | 1.38 (1.06-1.78); 0.015 | 1.14 (0.60-2.17); 0.684 | 1.21 (0.61-2.40); 0.581 | 0.76 (0.44-1.31); 0.324 | 0.73 (0.45-1.19); 0.210 | 1.00 (0.54-1.85); 0.999 | 0.74 (0.35-1.56); 0.430 | 1.16 (0.69-1.96); 0.583 |
| Sport | 0.85 (0.62-1.16); 0.313 | 0.96 (0.36-2.58); 0.942 | 0.69 (0.21-2.26); 0.543 | 0.75 (0.33-1.69); 0.486 | 0.43 (0.18-1.03); 0.057 | 1.09 (0.52-2.28); 0.829 | 0.66 (0.26-1.66); 0.374 | 1.22 (0.56-2.63); 0.615 |
| Psychology | 1.00 (0.73-1.37); 0.990 | 0.48 (0.14-1.63); 0.239 | 1.55 (0.71-3.35); 0.269 | 0.74 (0.33-1.67); 0.472 | 0.81 (0.43-1.52); 0.507 | 0.84 (0.40-1.77); 0.654 | 1.37 (0.73-2.58); 0.323 | 1.01 (0.51-2.00); 0.987 |
| Health | 0.93 (0.80-1.08); 0.340 | 0.97 (0.65-1.44); 0.868 | 1.01 (0.65-1.56); 0.976 | 0.93 (0.68-1.28); 0.666 | 1.03 (0.78-1.38); 0.817 | 0.92 (0.67-1.28); 0.638 | 0.66 (0.44-1.01); 0.056 | 1.19 (0.87-1.64); 0.278 |
| Responsible for buying food (no was the referent group) | | | | | | | | |
| Yes | 0.80 (0.68-0.94); 0.007 | 0.93 (0.61-1.42); 0.734 | 1.12 (0.71-1.76); 0.620 | 1.04 (0.73-1.47); 0.844 | 1.12 (0.81-1.55); 0.492 | 0.98 (0.68-1.39); 0.900 | 1.42 (0.96-2.09); 0.077 | 0.88 (0.62-1.24); 0.462 |
| Exercise (no was the referent group) | | | | | | | | |
| Yes | 0.76 (0.68-0.85); <0.001 | 0.75 (0.56-1.01); 0.055 | 0.87 (0.63-1.20); 0.391 | 0.88 (0.70-1.11); 0.272 | 1.18 (0.96-1.46); 0.124 | 1.00 (0.78-1.28); 0.996 | 0.86 (0.65-1.15); 0.311 | 0.86 (0.68-1.08); 0.193 |
| TV/computer, (hours/day) | 0.99 (0.97-1.01); 0.483 | 1.02 (0.97-1.07); 0.419 | 0.99 (0.93-1.04); 0.591 | 1.02 (0.98-1.06); 0.270 | 0.99 (0.95-1.02); 0.454 | 1.02 (0.98-1.06); 0.320 | 1.02 (0.97-1.07); 0.457 | 1.03 (1.00-1.07); 0.086 |
| Adherence to balanced/healthy diet (never/rarely was the referent group) | | | | | | | | |
| Often/always | 0.95 (0.82-1.10); 0.483 | 0.98 (0.68-1.42); 0.912 | 1.09 (0.72-1.63); 0.686 | 1.09 (0.80-1.49); 0.586 | 0.60 (0.46-0.78); <0.001 | 0.85 (0.62-1.16); 0.303 | 0.78 (0.53-1.14); 0.195 | 0.73 (0.56-0.96); 0.024 |
| EMFC | | | | | | | | |
| Cope with stress | 1.31 (1.14-1.51); <0.001 | 1.28 (0.89-1.84); 0.188 | 1.09 (0.72-1.64); 0.685 | 0.86 (0.64-1.15); 0.306 | 0.90 (0.69-1.18); 0.442 | 1.04 (0.76-1.43); 0.794 | 0.97 (0.68-1.39); 0.874 | 1.18 (0.89-1.58); 0.256 |
| Depression | 0.89 (0.77-1.02); 0.096 | 0.95 (0.65-1.40); 0.801 | 1.06 (0.70-1.61); 0.770 | 1.09 (0.81-1.48); 0.565 | 0.99 (0.76-1.30); 0.949 | 1.10 (0.81-1.50); 0.534 | 1.05 (0.74-1.49); 0.797 | 1.03 (0.78-1.38); 0.817 |
| Boredom | 1.11 (0.97-1.26); 0.138 | 1.18 (0.81-1.71); 0.383 | 1.08 (0.73-1.60); 0.706 | 1.23 (0.94-1.63); 0.137 | 0.96 (0.75-1.25); 0.784 | 0.86 (0.64-1.16); 0.327 | 0.89 (0.64-1.24); 0.502 | 0.87 (0.66-1.14); 0.314 |
| Loneliness | 1.12 (0.94-1.33); 0.221 | 0.82 (0.51-1.31); 0.403 | 1.13 (0.67-1.89); 0.650 | 0.84 (0.57-1.25); 0.392 | 0.90 (0.62-1.28); 0.548 | 1.06 (0.71-1.57); 0.787 | 0.84 (0.52-1.37); 0.491 | 0.86 (0.61-1.21); 0.376 |
| Emotional consolation | 1.22 (1.03-1.44); 0.020 | 1.40 (0.92-2.15); 0.121 | 0.99 (0.61-1.62); 0.981 | 1.21 (0.85-1.71); 0.291 | 0.82 (0.59-1.13); 0.229 | 1.03 (0.70-1.50); 0.896 | 1.02 (0.65-1.59); 0.941 | 1.47 (1.06-2.06); 0.022 |
| Helps control weight | 0.83 (0.73-0.93); 0.002 | 1.43 (1.04-1.96); 0.026 | 1.16 (0.82-1.64); 0.394 | 1.09 (0.85-1.40); 0.479 | 1.03 (0.82-1.29); 0.796 | 0.77 (0.59-1.02); 0.067 | 1.13 (0.83-1.53); 0.447 | 0.68 (0.52-0.88); 0.004 |
| Keeps awake and alert | 0.96 (0.85-1.09); 0.536 | 1.03 (0.74-1.45); 0.856 | 0.99 (0.69-1.41); 0.959 | 0.96 (0.74-1.26); 0.783 | 0.68 (0.53-0.87); 0.002 | 1.03 (0.78-1.35); 0.840 | 0.93 (0.68-1.28); 0.659 | 0.85 (0.66-1.09); 0.206 |
| Relaxation | 0.82 (0.73-0.93); 0.002 | 1.66 (1.21-2.28); 0.002 | 0.74 (0.52-1.06); 0.106 | 1.26 (0.99-1.60); 0.063 | 1.27 (1.02-1.58); 0.034 | 0.96 (0.73-1.25); 0.747 | 1.26 (0.93-1.69); 0.132 | 0.82 (0.63-1.06); 0.122 |
| Makes one feel good | 0.72 (0.63-0.81); <0.001 | 0.96 (0.70-1.33); 0.819 | 0.82 (0.59-1.15); 0.246 | 1.04 (0.81-1.33); 0.775 | 1.15 (0.92-1.44); 0.218 | 0.70 (0.54-0.91); 0.008 | 0.89 (0.66-1.21); 0.462 | 0.76 (0.59-0.99); 0.041 |

*(Continued)*

**Table 3.** (Continued)

BMI (< 25.0 kg/m² was the referent group)

| | BMI ≥ 25.0 kg/m² | Cardiovascular diseases | Diabetes mellitus | Hypercholesterolemia | Hypertension | Gastric disorders | Intestinal disorders | Obesity |
|---|---|---|---|---|---|---|---|---|
| BMI ≥25.0kg/m² | – | 1.16 (0.85-1.59); 0.341 | 1.67 (1.19-2.36); 0.003 | 1.05 (0.82-1.35); 0.677 | 1.67 (1.34-2.09); <0.001 | 0.93 (0.70-1.22); 0.587 | 0.77 (0.56-1.07); 0.117 | 18.67 (12.81-27.21); <0.001 |
| **Chronic noncommunicable diseases** | | | | | | | | |
| Cardiovascular diseases | 0.96 (0.70-1.32); 0.808 | – | 1.95 (1.22-3.10); 0.005 | 1.90 (1.33-2.71); <0.001 | 2.02 (1.44-2.83); <0.001 | 1.41 (0.84-2.34); 0.191 | 0.95 (0.43-2.06); 0.887 | 2.18 (1.45-3.28); <0.001 |
| Diabetes mellitus | 1.58 (1.10-2.25); 0.012 | 1.88 (1.16-3.04); 0.010 | – | 1.20 (0.76-1.88); 0.430 | 1.40 (0.94-2.08); 0.098 | 1.21 (0.65-2.24); 0.549 | 1.50 (0.74-3.03); 0.256 | 2.02 (1.31-3.12); 0.001 |
| Hypercholesterolemia | 0.83 (0.66-1.05); 0.126 | 2.06 (1.45-2.93); <0.001 | 1.31 (0.84-2.04); 0.227 | – | 2.99 (2.34-3.81); <0.001 | 2.42 (1.64-3.56); <0.001 | 0.63 (0.32-1.22); 0.173 | 1.62 (1.13-2.32); 0.009 |
| Hypertension | 1.40 (1.13-1.73); 0.002 | 2.16 (1.55-3.01); <0.001 | 1.45 (0.98-2.14); 0.060 | 2.91 (2.27-3.72); <0.001 | – | 1.28 (0.85-1.93); 0.240 | 0.54 (0.29-1.00); 0.049 | 1.85 (1.36-2.52); <0.001 |
| Gastric disorders | 0.96 (0.73-1.27); 0.779 | 1.45 (0.84-2.50); 0.186 | 1.18 (0.63-2.23); 0.608 | 2.59 (1.75-3.85); <0.001 | 1.28 (0.82-1.98); 0.273 | – | 5.81 (3.99-8.47); <0.001 | 1.81 (1.16-2.85); 0.010 |
| Intestinal disorders | 0.78 (0.57-1.08); 0.141 | 0.81 (0.35-1.86); 0.613 | 1.40 (0.67-2.90); 0.370 | 0.67 (0.34-1.32); 0.247 | 0.53 (0.28-1.02); 0.058 | 5.72 (3.92-8.34); <0.001 | – | 1.27 (0.70-2.33); 0.431 |
| Obesity | 17.89 (2.29-26.03); <0.001 | 2.10 (1.38-3.19); 0.001 | 2.14 (1.41-3.25); <0.001 | 1.91 (1.33 2.73); <0.001 | 2.36 (1.73-3.21); <0.001 | 1.79 (1.17-2.74); 0.007 | 1.49 (0.85-2.60); 0.162 | – |
| Other diseases | 1.43 (1.17-1.74); <0.001 | 0.95 (0.54-1.66); 0.858 | 0.67 (0.36-1.25); 0.205 | 0.60 (0.37-0.98); 0.040 | 0.81 (0.55-1.18); 0.272 | 1.15 (0.74-1.79); 0.527 | 1.17 (0.73-1.88); 0.508 | 0.73 (0.48-1.12); 0.154 |
| MHB | 0.98 (0.97-0.99); 0.005 | 0.98 (0.96-1.01); 0.294 | 1.00 (0.97-1.03); 0.918 | 1.00 (0.97-1.02); 0.693 | 1.02 (1.00-1.04); 0.063 | 0.98 (0.96-1.01); 0.185 | 1.04 (1.01-1.07); 0.017 | 0.96 (0.94-0.99); 0.001 |
| EMFC | 1.05 (1.03-1.07); <0.001 | 0.97 (0.93-1.02); 0.246 | 1.00 (0.96-1.05); 0.841 | 0.98 (0.95-1.02); 0.349 | 1.01 (0.98-1.04); 0.643 | 1.00 (0.97-1.04); 0.862 | 1.00 (0.96-1.04); 0.908 | 1.07 (1.03-1.12); <0.001 |

Note: N = number of participants; BMI = Body mass index; MHB = motivation for health behavior; EMFC = emotional motivation for food consumption; OR = odds ratio; 95% CI = 95% confidence interval.

In comparison to Portugal, higher odds of having a higher BMI were observed in all other European countries (p = 0.001 for Italy; p < 0.001 for other countries). Most European countries had lower odds of hypercholesterolemia and hypertension, but higher odds of CVDs, obesity, GDs, and IDs compared to the Portuguese population.

Older age was associated with a higher BMI [odds ratio (OR) = 1.05; 95% confidence interval (CI) 1.04–1.05; p < 0.001] and with most NCDs. Participants responsible for buying food [OR = 0.80 (95% CI 0.68–0.94); p = 0.007], those who exercised [OR = 0.76 (95% CI 0.68–0.85); p < 0.001], and those who consumed food to help control weight [OR = 0.83 (95% CI 0.73–0.93); p = 0.002], promote relaxation [OR = 0.82 (95% CI 0.73–0.93); p = 0.002], or make one feel good [OR = 0.72 (95% CI 0.63–0.81); p < 0.001] had lower odds of having a higher BMI. Conversely, participants who ate food as a coping mechanism for stress [OR = 1.31 (95% CI 1.14–1.51); p < 0.001] or for emotional consolation [OR = 1.22 (95% CI 1.03–1.44); p = 0.020] had higher odds of having a BMI ≥ 25.0 kg/m$^2$.

Participants with a BMI ≥ 25.0 kg/m$^2$ were more likely to have DM [OR = 1.67 (95% CI 1.19–2.36); p = 0.003], hypertension [OR = 1.67 (95% CI 1.34–2.09); p < 0.001], and to develop obesity [OR = 18.67 (95% CI 12.81–27.21); p < 0.001]. Participants with DM [OR = 1.58 (95% CI 1.10–2.25); p = 0.012], hypertension [OR = 1.40 (95% CI 1.13–1.73); p = 0.002], and other non-specific diseases [OR = 1.43 (95% CI 1.17–1.74); p < 0.001] were more likely to have a higher BMI. Participants with obesity were more likely to experience an additional increase in BMI [OR = 17.89 (95% CI 2.29–26.03); p < 0.001] and had a higher risk of developing most NCDs.

Compared to the Portuguese, Croats, Latvians, Lithuanians, Hungarians, and Romanians had higher odds of developing CVDs. Higher odds of CVDs were observed in participants with DM [OR = 1.88 (95% CI 1.16–3.04); p = 0.010], hypercholesterolemia [OR = 2.06 (95% CI 1.45–2.93); p < 0.001], hypertension [OR = 2.16 (95% CI 1.55–3.01); p < 0.001], and obesity [OR = 2.10 (95% CI 1.38–3.19); p = 0.001]. Participants with a higher risk of CVDs were those who consumed foods that promote relaxation [OR = 1.66 (95% CI 1.21–2.28); p = 0.002] and weight control [OR = 1.43 (95% CI 1.04–1.96); p = 0.026].

Compared to the Portuguese, Serbians [OR = 5.11 (95% CI 2.55–10.26); p < 0.001], Hungarians [OR = 4.15 (95% CI 2.26–7.62); p < 0.001], Slovenians [OR = 3.40 (95% CI 1.87–6.18); p < 0.001], Polish [OR = 3.33 (95% CI 1.75–6.33); p < 0.001], and Romanians [OR = 2.77 (95% CI 1.50–5.12); p = 0.001] had higher odds of obesity.

Higher odds of obesity were observed in participants who ate food for emotional consolation [OR = 1.47 (95% CI 1.06–2.06); p = 0.022], as well as in those with CVDs [OR = 2.18 (95% CI 1.45–3.28); p < 0.001], DM [OR = 2.02 (95% CI 1.31–3.12); p = 0.001], hypertension [OR = 1.85 (95% CI 1.36–2.52); p < 0.001], hypercholesterolemia [OR = 1.62 (95% CI 1.13–2.32); p = 0.009], and GDs [OR = 1.81 (95% CI 1.16–2.85); p = 0.010].

Participants with higher MHB had lower odds of having a higher BMI [OR = 0.98 (95% CI 0.97–0.99); p = 0.005] and obesity [OR = 0.96 (95% CI 0.94–0.99); p = 0.001], while participants with higher EMFC had a greater risk of having a higher BMI [OR = 1.05 (95% CI 1.03–1.07); p < 0.001] and developing obesity [OR = 1.07 (95% CI 1.03–1.12); p < 0.001] (Table 3).

## Discussion

This study aimed to explore the relationships between increased BMI, EMFC, MHB, and NCDs. The findings revealed significant associations between BMI, MHB, EMFC, and NCDs, including DM, hypertension, and obesity. Using food as a coping mechanism for stress and seeking emotional consolation through food were linked to an increased risk of a higher BMI. Conversely, individuals who consumed food for purposes such as weight management and well-being exhibited a lower likelihood of having a higher BMI and developing obesity. Although no direct associations were found between MHB and EMFC, both factors were linked to BMI, with MHB being negatively correlated and EMFC positively correlated.

Our findings are consistent with previous studies showing an association between BMI, EMFC, and the risk of NCDs. For example, one of the most recent clinical studies on the relationship between emotional eating, overweight/obesity, depression, anxiety/stress, and dietary habits found similar associations [24]. Psychological stress, depression, and

anxiety increase the risk of emotional eating, which favors high-energy but nutrient-poor foods [24]. Additionally, a study on the relationship between eating behavior and the severity of metabolic syndrome in young adults found that emotional eating is associated with greater severity of metabolic syndrome and, consequently, with worsening cardiometabolic health [25]. In contrast to previous studies, our study specifically examines the relationship between BMI, EMFC, and MHB, shedding light on the influence of these factors on the development of NCDs. These insights support the implementation of preventive measures by strengthening an individual's MHB as an effective tool for disease prevention.

The association between EMFC and BMI may contribute to the development of certain NCDs. It is well documented that a higher BMI is generally correlated with an increased susceptibility to various health conditions, including DM, heart disease, and certain types of cancer [1,13,26,27]. Other studies investigating the relationship between BMI and diseases such as DM, hypertension, and obesity have confirmed the associations found in our study [28,29]. Even within the normal BMI range, a higher BMI was associated with an increased risk of developing diseases [29]. The prevalence of DM, hypertension, and dyslipidemia increased with the severity of overweight and obesity [28]. The risk of DM and hypertension was significantly higher in individuals with a higher BMI compared to those with normal weight [28]. However, further research should prioritize exploring the neurological and genetic factors contributing to higher BMI and obesity. Existing literature highlights that genetics, biological mechanisms, and pathophysiological processes significantly influence differences in BMI, MHB, and EMFC. These differences are not merely the result of behavioral factors such as poor decision-making or lack of effort but rather reflect complex interactions between inherited traits and physiological processes [9,10]. Obesity has been linked to genetic predispositions that alter insulin and leptin signaling, contributing to insulin resistance and leading to behavioral disorders, including aggression and social isolation [9]. The discovery of a link between obesity and behavioral traits such as anxiety, aggression, autism, and depression has provided new insights into the development of obesity [9].

Emotional states such as perceived stress, loneliness, and the consumption of unhealthy foods for emotional comfort have been implicated in the development of obesity, related NCDs, and mental health disorders [24,30–32]. Emotions significantly influence food consumption patterns. Recent research indicates that stress and negative emotions can reduce the desire to eat, while increasing the consumption of calorie-dense foods [30]. It is widely known that many individuals turn to food for comfort or to cope with emotional distress, even when not hungry, a trend also supported by our study [16,31]. The present research revealed that individuals who resort to food for stress relief or emotional comfort had an increased likelihood of having a higher BMI, with emotional eating being positively associated with obesity. These results have been confirmed in other studies that have found a link between emotions, eating behavior, and obesity [24,33,34]. This underscores the significant impact of stress-coping mechanisms, such as emotional expression, seeking emotional support, and indulging in unhealthy foods during emotional states, on weight gain leading to overweight [30,35]. Conversely, participants who consumed foods that promote weight control, relaxation, and positive emotions demonstrated a lower risk of having a higher BMI. Other studies have also highlighted the effectiveness of relaxation techniques in weight management and overall well-being [36]. Furthermore, building stress resistance, along with a combination of a balanced diet and psychotherapy, can help individuals with emotional eating reduce the urge to eat in response to emotions [34,37]. These approaches can serve as useful strategies for managing weight gain.

MHB can exert a significant influence on both food consumption and BMI [15]. Individuals motivated to sustain a healthy weight and reduce their susceptibility to NCDs tend to adopt healthy eating patterns and engage in regular exercise. Their motivation may be influenced by various factors, including personal beliefs, social support, and access to resources such as healthy food and fitness facilities. Along with these factors, differences in MHB and EMFC could be significantly influenced by genetics, biology, and pathophysiology [9,10]. However, while differences in MHB and EMFC may be significantly influenced by genes, biology, and pathophysiology, it is also important to consider the interplay of environmental, behavioral, and psychosocial factors. For example, environmental influences such as stress, diet, and physical activity can reinforce the expression of genetic predispositions and biological signaling pathways [38]. Behavioral factors,

such as coping mechanisms and lifestyle choices, can either exacerbate or mitigate the effects of these intrinsic factors. Additionally, psychosocial influences, including social support, cultural norms, and socioeconomic status, can mediate the relationship between genetic and biological determinants and their phenotypic outcomes [38,39]. These complex interactions highlight the importance of understanding MHB and EMFC in the context of the integration of genetic, biological, and contextual factors. Given the intricate and multifaceted nature of these dynamics, comprehending them is crucial for individuals to make informed choices concerning their health and overall well-being [7,15].

The present study showed a higher likelihood of having a normal BMI in people who exercise regularly. It is widely recognized that increased physical activity correlates with lower body fat and BMI, while sedentary behaviors often lead to the opposite effect [40,41]. Therefore, exercise, along with healthy eating, plays a pivotal role in preventing NCDs, particularly in individuals with a stronger genetic predisposition [9,10,42]. Even in the absence of weight loss, exercise has the potential to mitigate the negative consequences of obesity on health [40]. Moreover, our study found a positive correlation between exercise and MHB, along with an inverse correlation with BMI, while healthy and balanced food intake was positively associated with MHB. Therefore, adopting a physically active lifestyle is essential for both primary and secondary prevention strategies against overweight and obesity [41,43,44].

Overweight and obesity are influenced by various environmental factors [45]. The present study revealed the highest BMI among participants from Hungary, Latvia, and Romania, while Portuguese participants exhibited the lowest BMI. Portuguese culture places great value on social relationships and family, supporting and encouraging one another to follow their passions, which can impact health behaviors in terms of diet and physical activity [46]. The diverse cultural, social, and economic factors across the European countries included in the study inevitably affect health behaviors and may have influenced the results we obtained. For example, Southern European countries often emphasize social gatherings and the enjoyment of food, which could influence dietary choices. Conversely, Nordic countries prioritize physical activity, outdoor recreation, and nature exploration as integral parts of leisure activities, shaping health behaviors related to exercise [47].

Choosing and eating healthy foods is a decisive factor in determining BMI and vice versa [48]. Participants who were responsible for buying food and who also exercised more had a lower likelihood of increasing their BMI. A balanced diet rich in fruits, vegetables, whole grains, lean proteins, and healthy fats is pivotal for maintaining a healthy weight and reducing the risk of NCDs and/or a higher BMI [49]. Conversely, diets rich in processed foods, sugars, and saturated fats can contribute to weight gain and the development of health problems [50]. Individuals who consumed appropriate amounts of nutrient-dense foods were better at controlling their weight, felt better, and were therefore less likely to increase their BMI [51,52].

Studies consistently confirm the causal link between obesity and numerous NCDs [53]. NCDs pose a significant health challenge for many countries, predominantly influenced by lifestyle factors, but also by genetic, biological, and pathophysiological mechanisms [9,10,53]. While the burden of NCDs varies across European countries, common trends and challenges persist [54]. According to the WHO, NCDs account for 78% of all deaths, with 85% of these deaths being premature [54,55]. However, as already mentioned, approximately 80% of NCDs [53] could be prevented by adopting positive lifestyle changes, including regular physical activity, adherence to a healthy dietary pattern, and abstaining from smoking, particularly in those predisposed to develop NCDs [5].

Across the countries included in this study, the proportion of NCDs among all deaths ranged from 86% to 95% [55]. However, efforts to combat NCDs vary across these nations. For instance, Lithuania has implemented measures to promote healthy lifestyles and improve healthcare access, while Romania has initiated campaigns to reduce smoking rates and encourage physical activity [54]. The Portuguese government has implemented various programs and initiatives to promote physical activity, healthy eating, and the prevention of overweight and obesity. The main focus of these initiatives is to create and enforce laws, regulations, and measures to improve food quality, reduce fat, sugar, and salt consumption, promote healthy eating patterns in communities, businesses, and schools, and increase physical activity [56].

Different socioeconomic and sociodemographic factors, including age, education level, income, and employment status, contribute to higher odds of obesity [5,20,27,43–45]. For instance, as individuals age, their body composition undergoes changes characterized by increased fat mass and redistribution [57]. Consistent with this, the findings of the present study indicate a positive association between BMI and age, suggesting that older adults are more prone to having a higher BMI. Furthermore, participants with a higher BMI were more likely to be male, reside in rural areas, have a lower educational level, be employed, and be married. These sociodemographic characteristics may contribute to an unhealthy lifestyle, including reduced physical activity, poor dietary choices, stress, inadequate sleep, feelings of loneliness, and social isolation [30,58].

Breaking the intricate interplay between BMI, emotional eating, and NCDs requires a comprehensive approach that addresses both physical and emotional aspects. Identifying triggers for emotional eating and developing healthier coping mechanisms are crucial steps in this process. Prioritizing whole foods, such as fruits, vegetables, whole grains, lean proteins, and healthy fats, forms the cornerstone of dietary intervention. Planning meals in advance and experimenting with new, nutritious recipes can add variety and enjoyment to eating habits. However, breaking this cycle can be challenging, highlighting the importance of seeking support from friends, family, healthcare professionals, or support groups. Collaborating with a registered dietitian can also provide valuable guidance and accountability.

This study expands the literature by providing large-scale, cross-national evidence on the relationship between BMI, EMFC, and health outcomes, demonstrating that emotional eating is a significant predictor of obesity and related NCDs. Emotional eating often plays a significant role in the onset and maintenance of NCDs, particularly among individuals struggling with weight management due to genetic, psychological, or emotional factors. Individuals with higher MHB tend to follow healthier dietary patterns, engage in regular exercise, and maintain a healthy weight, thereby reducing their susceptibility to NCDs. Furthermore, this study underscores the importance of lifestyle modification and psychological support in managing weight gain and preventing comorbidities.

Developing strong MHB, along with resilience and support, is crucial for maintaining a healthy lifestyle, achieving optimal weight, and preventing obesity and other NCDs [15,34,37]. This can be achieved through a combination of education, awareness, and access to resources such as healthy foods, physical activity programs, and counseling or coaching to facilitate positive behavioral changes. This study supports the integration of psychological support with lifestyle interventions to enhance MHB, combat emotional eating and weight gain, and improve the clinical management of NCDs. Considering the role of emotional regulation in eating behavior and implementing tailored interventions that combine psychological support with nutritional counseling may lead to more personalized, holistic treatment approaches [29,33,34,37]. Incorporating these findings into clinical practice could enable healthcare providers to offer more comprehensive strategies for obesity treatment and prevention, ultimately improving long-term patient outcomes. Future research should further investigate how emotional, sociodemographic, and psychological factors influence MHB, weight gain risk, and the development of NCDs. The results of this study contribute to shaping public health policy by providing evidence for integrated interventions that address lifestyle, psychological, and behavioral factors in the prevention and treatment of obesity and related diseases.

## Limitations

Despite the significance of this study in informing preventive measures for obesity and other NCDs, several limitations warrant consideration. Firstly, as a cross-sectional study, it cannot establish causal relationships between BMI, MHB, EMFC, unhealthy lifestyles, NCD incidence, and their associations. Secondly, anthropometric measurements were not conducted using standardized procedures; instead, participants self-reported their body weight and height, potentially introducing recall bias. Thirdly, additional measurements to differentiate between lean body mass, fat mass, and fat distribution were not performed. Fourthly, participants were not subject to clinical examinations or medical assessments to confirm their health conditions or genetic and biological predispositions. Lastly, variations

in lifestyle across European countries, disparities among study groups, and environmental differences may have influenced the results. Nonetheless, despite these limitations, this study makes a valuable scientific contribution to NCD prevention, particularly in addressing the multifaceted challenges posed by obesity and its associated health complications.

## Future directions

Future research should include longitudinal studies and randomized controlled trials to better understand the causal relationships between MHB, EMFC, BMI, and health outcomes. In addition, the role of genetic predispositions and the effectiveness of MHB and lifestyle interventions in preventing health impairments must not be neglected. It is recommended that all European countries prioritize investment in public health strategies to promote healthy lifestyles and improve access to nutritious food and other essential health resources.

Individuals struggling with emotional eating may benefit from professional support, including healthcare providers, therapists, and behavioral specialists who can offer guidance on increasing physical activity, managing emotional eating, and developing healthier coping mechanisms. Evidence-based approaches such as cognitive behavioral therapy and mindfulness-based interventions can help individuals develop a healthier relationship with food and manage emotional distress more effectively. Additionally, public health initiatives are urgently needed to address unhealthy dietary behaviors associated with emotional states.

## Conclusions

This study contributes to the understanding of human eating behavior by considering psychological factors. While no associations were found between MHB and EMFC, both were linked to BMI - MHB negatively and EMFC positively. Targeted preventive measures are essential to reduce the health risks of emotional eating and the resulting obesity as a complex multifactorial disorder associated with various health risk factors. Effective interventions should focus on dietary modifications, self-regulation, resilience-building, education on adaptive coping strategies, and promoting regular physical activity.

## Supporting information

**S1 Table. Inclusivity in global research.**
(DOCX)

**S2 Table. Sociodemographic characteristics of European population (N = 9,036).**
(DOCX)

## Acknowledgments

The authors would like to thank the University of Zadar and CERNAS Research Center, and the Polytechnic Institute of Viseu for their support. This work was prepared in the ambit of the multinational project EATMOT from the CI and DETS Research Center (Polytechnic Institute of Viseu, Portugal) with reference PROJ/CI and DETS/CGD/0012.

## Author contributions

**Conceptualization:** Marija Ljubičić, Marijana Matek Sarić, Ana Sarić.

**Data curation:** Marija Ljubičić, Marijana Matek Sarić.

**Formal analysis:** Marija Ljubičić, Marijana Matek Sarić.

**Funding acquisition:** Marijana Matek Sarić, Raquel Guiné.

**Investigation:** Marija Ljubičić, Marijana Matek Sarić, Tamara Sorić, Ana Sarić, Ivo Klarin, Boris Dželalija, Alan Medić, Ivo Dilber, Ivana Rumbak, Jasmina Ranilović, Maria Papageorgiou, Viktória Szűcs, Elena Vittadini, Dace Klava, Lucia Frez Muñoz, Małgorzata Korzeniowska, Monica Tarcea, Ilija Djekić, Maša Černelič Bizjak, Raquel Guiné.

**Methodology:** Marija Ljubičić, Marijana Matek Sarić.

**Project administration:** Marijana Matek Sarić, Raquel Guiné.

**Resources:** Marijana Matek Sarić, Raquel Guiné.

**Software:** Marija Ljubičić, Marijana Matek Sarić, Ana Sarić.

**Supervision:** Marijana Matek Sarić, Raquel Guiné.

**Validation:** Marija Ljubičić, Marijana Matek Sarić.

**Visualization:** Marija Ljubičić, Marijana Matek Sarić, Tamara Sorić, Ana Sarić.

**Writing – original draft:** Marija Ljubičić, Marijana Matek Sarić, Ana Sarić.

**Writing – review & editing:** Marija Ljubičić, Marijana Matek Sarić, Tamara Sorić, Ana Sarić, Ivo Klarin, Boris Dželalija, Alan Medić, Ivo Dilber, Ivana Rumbak, Jasmina Ranilović, Maria Papageorgiou, Viktória Szűcs, Elena Vittadini, Dace Klava, Lucia Frez Muñoz, Małgorzata Korzeniowska, Monica Tarcea, Ilija Djekić, Maša Černelič Bizjak, Raquel Guiné.

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
