## [Decision Letter · Decision Letter 0]

20 Dec 2024

PONE-D-24-44681The interplay between body mass index, motivation for food consumption, and noncommunicable diseases in the European population: a cross-sectional studyPLOS ONE

Dear Dr. Matek Sarić,

Thank you for submitting your manuscript to PLOS ONE. After careful consideration, we feel that it has merit but does not fully meet PLOS ONE’s publication criteria as it currently stands. Therefore, we invite you to submit a revised version of the manuscript that addresses the points raised during the review process.

Please note that we have only been able to secure a single reviewer to assess your manuscript. We are issuing a decision on your manuscript at this point to prevent further delays in the evaluation of your manuscript. Please be aware that the editor who handles your revised manuscript might find it necessary to invite additional reviewers to assess this work once the revised manuscript is submitted. However, we will aim to proceed on the basis of this single review if possible.  Could you please revise the manuscript to carefully address the concerns raised? Please submit your revised manuscript by Feb 02 2025 11:59PM. If you will need more time than this to complete your revisions, please reply to this message or contact the journal office at plosone@plos.org . Please include the following items when submitting your revised manuscript:

We look forward to receiving your revised manuscript.

Kind regards,

Helen Howard

Staff Editor

PLOS ONE

Reviewers' comments:

Reviewer's Responses to Questions

**Comments to the Author**

1. Is the manuscript technically sound, and do the data support the conclusions?

Reviewer #1: Yes

2. Has the statistical analysis been performed appropriately and rigorously? 

Reviewer #1: Yes

3. Have the authors made all data underlying the findings in their manuscript fully available?

Reviewer #1: Yes

4. Is the manuscript presented in an intelligible fashion and written in standard English?

Reviewer #1: Yes

5. Review Comments to the Author

Reviewer #1: PONE-D-24-44681

This paper describes a large cross sectional cohort study of European adults, examining associations between self-reported emotional state, food intake, motivation and various non-communicable disease related outcomes. It is an excellent paper overall. I would have the following comments.

Firstly, in the introduction and throughout, it is clear the authors have a “behaviourist” view on obesity and why it affects some people more than others. For example, they state that “… maintaining health requires strong motivation combined with positive lifestyle habits such as eating healthy foods…”. I think that this could be disputed, because it infers that people with obesity lack motivation, or indeed that thin people must be highly motivated. I’m sure that this is not what the authors intend, but it would really strengthen the relevance and acceptability of the papers very interesting findings if the authors acknowledged more the very profound biological and genetic basis for variations in the complex neurobehavioural responses to what has become an increasingly obesogenic environment. Within that, the nuances of emotional state are critically important considerations, but I’d love to see more consideration of this issue throughout. Furthermore, I’d be careful of the use of the term “habits”. Arguably the strong biological basis for obesity makes the various implicated behaviour patterns more than mere habits. Perhaps consider that variations in MHB and EMFC could be strongly influenced by genes, biology and pathophysiology, rather than by affected individuals just not thinking in the right way or not making enough effort. I think this is really important for this paper, in order that it has its deserved impact. Consider the famous paper in 2018 by Rogers in the Lancet, “telling us what did not cause the obesity epidemic”.

It is not clear why the authors dichotomised BMI. Can I check are the regression analyses in table 2 based on the treatment of BMI as a continuous rather than a categorical variable? Also, the statement that “Participants with BMI ≥ 25.0 kg/m2 had an increased risk of… development of obesity” seems tautological (and inconsistent with the cross-sectional nature of the study). Of course those with a high BMI were more likely to have obesity.

Rather than “Association between body mass index and incidence of chronic noncommunicable diseases”, it should be prevalence (as this is cross-sectional). Again, the statement that “Participants with obesity had almost 18 times higher odds of additionally increasing their BMI” makes no sense to me.

I don’t think the between country comparisons offer much in the paper relevant to the hypothesis in hand. It is noteworthy though that 7.2% of Serbian adults have a BMI under 18.5 kg/m2.

6. PLOS authors have the option to publish the peer review history of their article (what does this mean? ). If published, this will include your full peer review and any attached files.

**Do you want your identity to be public for this peer review?** For information about this choice, including consent withdrawal, please see our Privacy Policy .

Reviewer #1: **Yes: ** Francis Finucane

---

## [Author Response · Author response to Decision Letter 1]

23 Jan 2025

Reviewer's Responses to Questions

1. Is the manuscript technically sound, and do the data support the conclusions?

Reviewer #1: Yes

Authors: Thank you for your response.

2. Has the statistical analysis been performed appropriately and rigorously?

Reviewer #1: Yes

Authors: Thank you for your response.

3. Have the authors made all data underlying the findings in their manuscript fully available?

Reviewer #1: Yes

Authors: Thank you for your response.

4. Is the manuscript presented in an intelligible fashion and written in standard English?

Reviewer #1: Yes

Authors: Thank you for your response.

5. Review Comments to the Author

Reviewer #1: PONE-D-24-44681

This paper describes a large cross sectional cohort study of European adults, examining associations between self-reported emotional state, food intake, motivation and various non-communicable disease related outcomes. It is an excellent paper overall.

Response: Thank you for your kind and encouraging feedback. We sincerely appreciate your recognition of our work and are delighted that you found the paper to be of high quality. Your insights and support mean a great deal to us.

I would have the following comments.

1. Firstly, in the introduction and throughout, it is clear the authors have a “behaviourist” view on obesity and why it affects some people more than others. For example, they state that “… maintaining health requires strong motivation combined with positive lifestyle habits such as eating healthy foods…”. I think that this could be disputed, because it infers that people with obesity lack motivation, or indeed that thin people must be highly motivated. I’m sure that this is not what the authors intend, but it would really strengthen the relevance and acceptability of the papers very interesting findings if the authors acknowledged more the very profound biological and genetic basis for variations in the complex neurobehavioural responses to what has become an increasingly obesogenic environment. Within that, the nuances of emotional state are critically important considerations, but I’d love to see more consideration of this issue throughout.

Response Ad 1. Thank you for your valuable comments. We deeply appreciate your thoughtful feedback and have carefully followed your guidance in revising the introduction and throughout the manuscript. Your suggestions have been thoughtfully considered to enhance the quality and clarity of our work.

Among other things, after the sentence“… maintaining health requires strong motivation combined with positive lifestyle patterns such as eating healthy foods…” we added the new sentence “Hence,, it is important to recognize that the development of obesity is not only due to individual behaviors or lack of motivation, as genetic, epigenetic, neurobiological and environmental factors also play an important role in weight gain and its effects on different individuals”

We have reworded the introduction in line with your suggestions and ensured that this approach is consistently maintained throughout the whole manuscript.

2. Furthermore, I’d be careful of the use of the term “habits”. Arguably the strong biological basis for obesity makes the various implicated behaviour patterns more than mere habits. Perhaps consider that variations in MHB and EMFC could be strongly influenced by genes, biology and pathophysiology, rather than by affected individuals just not thinking in the right way or not making enough effort. I think this is really important for this paper, in order that it has its deserved impact. Consider the famous paper in 2018 by Rogers in the Lancet, “telling us what did not cause the obesity epidemic”.

Response Ad 2. Thank you for this comment. We agree with you. We have replaced the term habit with a more appropriate term or corrected the sentences. We have also emphasized the importance of genetic, biological and pathophysiological mechanisms, as you recommended. We have done this throughout the manuscript. We have carefully reviewed the article by Rogers in the Lancet 2018 and other similar references and add them to the reference list.

3. It is not clear why the authors dichotomised BMI.

Response Ad 3. Thank you for your question regarding the dichotomization of BMI at a threshold of 25. This cutoff was chosen because it is a well-established and standardized criterion set by the World Health Organization (WHO) to distinguish between normal weight (BMI < 25) and overweight (BMI ≥ 25) in adults. Using this threshold ensures consistency with previous research and aligns with public health guidelines, making it easier to compare findings and apply them in both clinical and epidemiological settings. Additionally, dichotomizing BMI at this point helps structure the analysis of its relationship with other variables in the study. We have added this explanation to the statistical analysis section for clarity.

“

“BMI was dichotomized according to the World Health Organization (WHO) classification, as described in Study design and participants section, to ensure adherence to established public health standards and to facilitate the interpretation of results. However, it is important to note that a BMI of 25.0 kg/m2 or higher encompasses both overweight and obesity, with obesity defined as a BMI of 30.0 kg/m2 or greater. Conversely, a BMI below 25.0 kg/m2 does not necessarily indicate normal body weight. For instance, a BMI of <18.5 kg/m2, reflects malnutrition rather than a healthy weight. This highlights the need to interpret BMI values within the broader context of individual health and nutritional status. NCDs variables were also dichotomized (presence or absence of disease) to ensure interpretability within the logistic regression model. Predictors in all these models included the same variables as in the linear regression models mentioned above. Referent groups were also defined as in previous regression models. Beta values were converted into odds ratios with 95% confidence intervals (lower and upper bounds).

To minimize statistical bias, all predictors were simultaneously included in the regression models. In certain models, the primary outcome variables were also incorporated as predictors in the analysis.”

4. Can I check are the regression analyses in table 2 based on the treatment of BMI as a continuous rather than a categorical variable?

Response Ad 4. Table 2 shows a linear regression model, and BMI is included as a continuous outcome variable. We examined BMI as an outcome variable in its relationship with motivation. According to the principles of statistics, linear regression requires the dependent variable (outcome) to be continuous. This ensures the fulfillment of key model assumptions, including the linearity of relationships between variables. In linear regression BMI as outcome variable should be continuous and because of that we left it as continuous, while we dichotomized it in the logistic regression.

For clarity, we add the sentence in statistical method “BMI was included as a continuous variable in all liner regression models.”

5. Also, the statement that “Participants with BMI ≥ 25.0 kg/m2 had an increased risk of… development of obesity” seems tautological (and inconsistent with the cross-sectional nature of the study). Of course, those with a high BMI were more likely to have obesity.

Response Ad 5. Thank you for these comments. We agree that in the context of a cross-sectional study, the OR (Odds ratio) represents the probability of an association between an exposure and an outcome, rather than a measure of risk, as the study design does not establish causality. For clarity, we have followed your recommendation and rephrased “risk” to “likelihood” or “probability” of an association between an exposure and an outcome.

6. Rather than “Association between body mass index and incidence of chronic noncommunicable diseases”, it should be prevalence (as this is cross-sectional).

Response Ad 6. Thank you. We deleted the word “incidence”.

7. Again, the statement that “Participants with obesity had almost 18 times higher odds of additionally increasing their BMI” makes no sense to me.

Response Ad 7. We reformulated this sentence “Participants with obesity were more likely to have an additional increase in BMI [OR=17.89 (95% CI 2.29-26.03); p<0.001] and were at higher risk of developing most NCDs.”

8. I don’t think the between country comparisons offer much in the paper relevant to the hypothesis in hand. It is noteworthy though that 7.2% of Serbian adults have a BMI under 18.5 kg/m2.

Response Ad 8. Thank you for your thoughtful feedback. We appreciate your pointing out the limited relevance of country comparisons in the context of our hypothesis. Although these comparisons do not directly address the hypotheses, we included them to provide a broader context and explore possible national differences that might influence the relationships between BMI, lifestyle, food consumption, socioeconomic and emotional states. As you mentioned, the notable percentage of Serbian adults with a BMI below 18.5 kg/m2 is an interesting finding that could be explored further in the context of broader demographic trends. We have added these comparisons for Serbia. We also add the results from Portugal, where the prevalence of normal BMI is highest “Participants from Portugal had the highest proportion of people with a normal BMI (79.9%) compared to other countries, while 7.2% of Serbian adults had a BMI below 18.5 kg/m2.“

PLOS authors have the option to publish the peer review history of their article (what does this mean?). If published, this will include your full peer review and any attached files.

Do you want your identity to be public for this peer review? For information about this choice, including consent withdrawal, please see our Privacy Policy.

Reviewer #1: Yes: Francis Finucane

---

## [Decision Letter · Decision Letter 1]

11 Feb 2025

PONE-D-24-44681R1The interplay between body mass index, motivation for food consumption, and noncommunicable diseases in the European population: a cross-sectional studyPLOS ONE

Dear Dr. Matek Sarić,

Thank you for submitting your manuscript to PLOS ONE. After careful consideration, we feel that it has merit but does not fully meet PLOS ONE’s publication criteria as it currently stands. Therefore, we invite you to submit a revised version of the manuscript that addresses the points raised during the review process.

We look forward to receiving your revised manuscript.

Kind regards,

Leonardo Roever

Academic Editor

PLOS ONE

Journal Requirements:

Additional Editor Comments:

(PLACE INSERTS IN A DIFFERENT COLOR FONT TO IDENTIFY CHANGES IN THE ARTICLE)

PUT IN RED LETTERS - ANSWER THE QUESTIONS BELOW POINT BY POINT.

PLEASE INCLUDE ALL REQUESTS IN THE MANUSCRIPT.

Include in article

0 - Please correct grammatical and spelling errors

1 - Abstract

Conclusions: State only what your study found; do not include extraneous information not backed up by the results.

2 - Discussion

Compare and contrast your study with others in the most relevant world literature, particularly the recent literature.

3 - What new information is sufficient to modify existing clinical practice?

4 -What are the conclusions and implications for current practice, and particularly for future research that may have a significant impact on clinical decisions?

5 - How can this study affect public policies related to health?

6 - What does this study add to the literature?

7 – Improve - At the end of the Discussion, under the subheading "Limitations," review the limitations of your study.

8 - At the end of the limitations, under the subheading " Future directions".

9 - Conclusion

Take special care to draw your conclusions only from your results and verify that your conclusions are firmly supported by your data.

Reviewers' comments:

Reviewer's Responses to Questions

**Comments to the Author**

1. If the authors have adequately addressed your comments raised in a previous round of review and you feel that this manuscript is now acceptable for publication, you may indicate that here to bypass the “Comments to the Author” section, enter your conflict of interest statement in the “Confidential to Editor” section, and submit your "Accept" recommendation.

Reviewer #1: All comments have been addressed

Reviewer #2: All comments have been addressed

2. Is the manuscript technically sound, and do the data support the conclusions?

Reviewer #1: Yes

Reviewer #2: Yes

3. Has the statistical analysis been performed appropriately and rigorously? 

Reviewer #1: Yes

Reviewer #2: Yes

4. Have the authors made all data underlying the findings in their manuscript fully available?

Reviewer #1: Yes

Reviewer #2: Yes

5. Is the manuscript presented in an intelligible fashion and written in standard English?

Reviewer #1: Yes

Reviewer #2: Yes

6. Review Comments to the Author

Reviewer #1: Comments fully addressed thanks. I have no further comments to make. Congratulations on an excellent paper.

Reviewer #2: Redundancy in Certain Statements:

Example: “Participants with BMI ≥ 25 had an increased risk of obesity.”

This is tautological, as BMI ≥ 30 already defines obesity.

Suggested Revision: “Participants with a BMI ≥ 25 were more likely to be overweight or obese.”

Some Long or Complex Sentences Could Be Simplified for Readability:

A few sentences contain multiple clauses, making them harder to follow.

Example:

Original: “Maintaining health requires strong motivation combined with positive lifestyle patterns such as eating healthy foods, exercising regularly, controlling stress, and abstaining from smoking and alcohol consumption.”

Suggested Revision: “Maintaining health requires motivation and positive lifestyle choices, including a balanced diet, regular exercise, stress management, and avoiding smoking and alcohol.”

7. PLOS authors have the option to publish the peer review history of their article (what does this mean? ). If published, this will include your full peer review and any attached files.

**Do you want your identity to be public for this peer review?** For information about this choice, including consent withdrawal, please see our Privacy Policy .

Reviewer #1: **Yes: ** Francis Finucane

Reviewer #2: **Yes: ** Anees Alyafei

---

## [Author Response · Author response to Decision Letter 2]

12 Mar 2025

12nd March 2025

Dear Editor,

I hope this letter finds you well. On behalf of my co-authors, I am pleased to resubmit our manuscript, entitled "The interplay between body mass index, motivation for food consumption, and noncommunicable diseases in the European population: A cross-sectional study," for your kind consideration for publication in PLOS ONE.

We have carefully addressed all the comments provided by the reviewers and the editorial team and have revised the manuscript accordingly. The updated version reflects comprehensive improvements to enhance its clarity, rigor, and alignment with the high standards of your journal. For your convenience, we have included a detailed point-by-point response outlining the revisions made in response to the feedback received. Additionally, we have conducted a thorough grammatical and stylistic review of the entire manuscript.

This study examines the intricate relationships between body mass index (BMI), motivations for food consumption, and their associations with noncommunicable diseases across diverse European populations. We believe our findings offer valuable insights into public health and nutrition, with potential implications for policy-making and interventions aimed at reducing the burden of noncommunicable diseases.

We sincerely appreciate the opportunity to improve our manuscript based on the constructive feedback provided, and we hope the revised version meets the expectations of the reviewers and the editorial team. Thank you for your time and consideration. Please do not hesitate to contact us should any further clarifications or revisions be required.

We look forward to your response and hope our work will find a place within the pages of PLOS ONE.

Kind regards,

Professor Marijana Matek Sarić, Ph.D.

On behalf of my co-authors

---

## [Editor Report · Decision Letter 2]

14 Mar 2025

PONE-D-24-44681R2The interplay between body mass index, motivation for food consumption, and noncommunicable diseases in the European population: a cross-sectional studyPLOS ONE

Dear Dr. Matek Sarić,

Thank you for submitting your manuscript to PLOS ONE. After careful consideration, we feel that it has merit but does not fully meet PLOS ONE’s publication criteria as it currently stands. Therefore, we invite you to submit a revised version of the manuscript that addresses the points raised during the review process.

We look forward to receiving your revised manuscript.

Kind regards,

Leonardo Roever

Academic Editor

PLOS ONE

Journal Requirements:

Additional Editor Comments :

(PLACE INSERTS IN A DIFFERENT COLOR FONT TO IDENTIFY CHANGES IN THE ARTICLE)

PUT IN RED LETTERS - ANSWER THE QUESTIONS BELOW POINT BY POINT.

INCLUDE THE PAGE AND LINE WHERE YOU ARE MAKING THE CHANGE.

PLEASE INCLUDE ALL REQUESTS IN THE MANUSCRIPT.

Include in article

1 - Rephrase the conclusion.

Include only what your data shows.

The third paragraph of the conclusion should be in future directions as it is not a conclusion that your data shows.

---

## [Author Response · Author response to Decision Letter 3]

20 Mar 2025

PONE-D-24-44681R2 - [EMID:cc0bfa0781d4c25a]

The interplay between body mass index, motivation for food consumption, and noncommunicable diseases in the European population: a cross-sectional study

Point 0. Please review your reference list to ensure that it is complete and correct. If you have cited papers that have been retracted, please include the rationale for doing so in the manuscript text, or remove these references and replace them with relevant current references. Any changes to the reference list should be mentioned in the rebuttal letter that accompanies your revised manuscript. If you need to cite a retracted article, indicate the article’s retracted status in the References list and also include a citation and full reference for the retraction notice.

Response Ad 0. Thank you for your comment.

We have reviewed our reference list to ensure it is complete and accurate. We have not cited any retracted papers. The changes we have made are as follows:

• Reference 2: "Organisation" has been changed to "Organization."

• Reference 3: "2020;17:1–10" has been changed to "2020, 17:8878."

• Reference 5: In the journal title, the abbreviation "Heal" has been changed to "Health."

• Reference 6: The article and page number were missing and have now been updated to "2020, 22: e17640."

• Reference 7: The DOI was incorrect; the unnecessary "/S1" has been removed.

• Reference 8: The article and page number were missing and have now been updated to "2020, 20:1165." The DOI was also incorrect as it contained a reference to "/TABLES/3."

• Reference 10: The journal abbreviation was changed from "Lancet Public Heal" to "Lancet Public Health."

• Reference 12: Instead of just "559," the range "559-66" has been added.

• Reference 14: The year was incorrect and has now been changed to 2025.

• Reference 15: "Sarić MM" has been changed to "Matek Sarić M"; similarly, "Barić IC" is now written as "Colić Barić I."

• Reference 16: The year was "2022"; the volume and article are now correctly written as "2022; 2022:4243868." Additionally, "de la" is now correctly capitalized in the author's name.

• Reference 18: "Ridder D de" has been changed to "de Ridder D." The duplicate DOI has been removed, and the journal name "Psychol Health" has been added.

• Reference 21: The DOI has been added: https://www.researchgate.net/publication/321142030.

• Reference 22: "Guiné Raquel PF" has been changed to "Guiné RPF." The journal abbreviation has been changed from "Zdr. Varst" to "Slov J Public Heal," and page numbers "4-9" have been added.

• Reference 23: "Guine" has been changed to "Guiné." The third author's name has been corrected to "Szűcs V," and the fifth and sixth authors' names have been corrected to "Ljubičić M, Černelić-Bizjak M." Additionally, "2020, Vol IX, Page 888" has been removed.

• Reference 26: Page numbers "944-53" have been added.

• Reference 28: The unnecessary "2023 131" has been removed; instead of "1-10," it now states "2346."

• Reference 29: "2024; 14: e083443" has been added.

• Reference 31: A space has been added between "Pilot" and "Study." Page numbers have been corrected to "131-9."

• Reference 33: This was a duplicate of Reference 24, so it has been removed, and the numbering in the list has been adjusted.

• Reference 34 (now 33): "/TABLES/2" has been removed.

• Reference 34: Instead of "1-9," it now states "129," and "/TABLES/3" has been removed.

• Reference 35: "215,10" has been corrected to "2015; 10: e0118105."

• Reference 42: "315" has been corrected to "315-33."

• Reference 43: The authors are now correctly listed as "You HW, Tan PL, Mat Ludin AF." The DOI has been added: https://doi.org/10.31436/imjm.v19i2.1567.

• Reference 44: "1-9" has been corrected to "1354," and "/TABLES/1" has been removed.

• Reference 46: "Cult Atlas" has been changed to "Cultural Atlas," as it was not an abbreviation.

• Reference 47: "2023; 23" has been changed to "2023:1514."

• Reference 48: "2022; 13" has been changed to "2022; 13:873835." The incorrect DOI has been removed and replaced with the correct one.

• Reference 50: The incorrect link has been replaced with the correct one: https://www.cdc.gov/healthy-weight-growth/healthy-eating/?CDC_AAref_Val=https://www.cdc.gov/healthyweight/healthy_eating/index.html. The article title has been changed to Tips for Healthy Eating for Healthy Weight 2023.

• Reference 51: Instead of "1-9," it now states "1990." The incorrect DOI and "/TABLES/5" have been removed.

• Reference 52: The volume was incorrectly listed as 31 but should be 43.

• Reference 53: "2021; 19" has been changed to "2021; 19:320."

• Reference 54: "World Heal Organ" has been changed to "WHO."

• Reference 56: The reference has been revised for clarity:

Before:

Ministry of Health of Portugal. “PNPAS” National Programme for the Promotion of Healthy Eating Portugal. Title in original language: Which “life stage(s)” for CVD prevention targets the intervention? What is the level of implementation of your example of good practice? n.d. www.chrodis.eu (accessed January 2, 2024).

After:

Ministry of Health of Portugal. “PNPAS” National Programme for the Promotion of Healthy Eating Portugal. n.d. www.chrodis.eu (accessed January 2, 2024).

• Reference 57: "2020, 21" has been changed to "2020, 21; e13035."

• References 60 and 61: Removed, as this part of the conclusion was deleted in the revised article.

Additional Editor Comments:

(PLACE INSERTS IN A DIFFERENT COLOR FONT TO IDENTIFY CHANGES IN THE ARTICLE)

PUT IN RED LETTERS - ANSWER THE QUESTIONS BELOW POINT BY POINT.

INCLUDE THE PAGE AND LINE WHERE YOU ARE MAKING THE CHANGE.

PLEASE INCLUDE ALL REQUESTS IN THE MANUSCRIPT.

We followed the instructions and acted accordingly.

Thank you.

Include in article

Point 1. Rephrase the conclusion.

Response Ad 1. Thank you for your comment. We reformulated the conclusion.

Point 2. Include only what your data shows.

Response Ad 2. Thank you for your comment. We have ensured that conclusion include only findings directly supported by our data.

Point 3. The third paragraph of the conclusion should be in future directions as it is not a conclusion that your data shows.

Response Ad 3. Thank you for your feedback. We have revised the conclusion and moved the third paragraph to the "Future directions" section to ensure that the conclusion reflects only the findings supported by our data.

---

## [Editor Report · Decision Letter 3]

24 Mar 2025

The interplay between body mass index, motivation for food consumption, and noncommunicable diseases in the European population: a cross-sectional study

PONE-D-24-44681R3

Dear Dr. Matek Sarić,

We’re pleased to inform you that your manuscript has been judged scientifically suitable for publication and will be formally accepted for publication once it meets all outstanding technical requirements.

Kind regards,

Leonardo Roever PhD, MBA

Academic Editor

PLOS ONE

---

## [Editor Report · Acceptance letter]

PONE-D-24-44681R3

PLOS ONE

Dear Dr. Matek Sarić,

I'm pleased to inform you that your manuscript has been deemed suitable for publication in PLOS ONE. Congratulations! Your manuscript is now being handed over to our production team.

Kind regards,

on behalf of

Prof. Dr. Leonardo Roever

Academic Editor

PLOS ONE